# Durable organic nonlinear optical membranes for thermotolerant lightings and in vivo bioimaging

Tian Tian[1,5], Yuxuan Fang[1,5], Wenhui Wang[1], Meifang Yang[1], Ying Tan[1], Chuan Xu[2], Shuo Zhang[1], Yuxin Chen[3], Mingyi Xu [4] ✉, Bin Cai [2] ✉ & Wu-Qiang Wu [1] ✉

Organic nonlinear optical materials have potential in applications such as lightings and bioimaging, but tend to have low photoluminescent quantum yields and are prone to lose the nonlinear optical activity. Herein, we demonstrate to weave large-area, flexible organic nonlinear optical membranes composed of 4-N,N-dimethylamino-4′-N′-methyl-stilbazolium tosylate@cyclodextrin host-guest supramolecular complex. These membranes exhibited a record high photoluminescence quantum yield of 73.5%, and could continuously emit orange luminescence even being heated at 300 °C, thus enabling the fabrication of thermotolerant light-emitting diodes. The nonlinear optical property of these membranes can be well-preserved even in polar environment. The supramolecular assemblies with multiphoton absorption characteristics were used for in vivo real-time imaging of Escherichia coli at 1000 nm excitation. These findings demonstrate to achieve scalable fabrication of organic nonlinear optical materials with high photoluminescence quantum yields, and good stability against thermal stress and polar environment for high-performance, durable optoelectronic devices and humanized multiphoton bio-probes.

Organic nonlinear optical (NLO) materials have attracted a great deal of research interest owing to their tremendous potential in a variety of applications such as microlasers, on-chip optical communication, lighting, displaying, biological sensing, and THz generation[1–4], etc. As a benchmark of NLO materials, 4-N,N-dimethylamino-4′-N′-methyl-stilbazolium tosylate (DAST) crystal is of particular interest due to its high electro-optic coefficient[5–7], high NLO susceptibilities, and good second-harmonic generation (SHG) property. DAST is a donor-π-acceptor (D-π-A)-type molecule, which has excellent two-photon or multiphoton absorption (2PA or MPA) characteristics that could emit visible light[8,9]. In addition, DAST is a highly polarized compound with a non-centrosymmetric arrangement and large dipole moment, and its energy level and excited-state processes are very sensitive to the molecular environments, which severely limits its availability and practical application[5,10]. First, the strict anhydrous atmosphere is of paramount importance for fabricating high-quality DAST crystals, otherwise, the molecular structure with a non-centrosymmetric arrangement will be destroyed and the NLO activity of materials will be lost[11–13]. Secondly, the DAST molecules suffer from notorious aggregation-caused quenching (ACQ) and other nonradiative decay channels like the intramolecular charge transfer (ICT) process, which largely limits the improvement of luminescent performance. Worse

[1]MOE Key Laboratory of Bioinorganic and Synthetic Chemistry, Lehn Institute of Functional Materials, School of Chemistry, Sun Yat-sen University, Guangzhou 510006, P. R. China. [2]Shanghai Key Lab of Modern Optical System, Ministry of Education, University of Shanghai for Science and Technology, Shanghai 200093, China. [3]Instrumental Analysis and Research Center, Sun Yat-sen University, Guangzhou 510275, P. R. China. [4]Guangdong Key Laboratory of Environmental Catalysis and Health Risk Control, School of Environmental Science and Engineering, Institute of Environmental Health and Pollution Control, Guangdong University of Technology, Guangzhou 510006, China. [5]These authors contributed equally: Tian Tian, Yuxuan Fang. ✉e-mail: anasyxmy@foxmail.com; bullcai@usst.edu.cn; wuwq36@mail.sysu.edu.cn

still, DAST crystals have a low photobleaching threshold under excitation by blue, green, and even near-infrared (NIR) light because of the occurrence of trans−cis isomerization of stilbazolium[10,11]. Besides, most DAST crystals grown on the rigid substrate are tiny, fragile, and easy to be destroyed during the peeling-off process, thus limiting their potential to be applied in integrated optical/photonic/optoelectronic devices.

Over the past two decades, several fluorescent and phosphorescent materials have been designed and applied in bioimaging[14–16]. However, most of the applicable bioimaging systems are based on inorganic materials such as transition metal chalcogenides quantum dots/nanocrystals (i.e., FeSe, ZnS, CdSe, CdTe, etc.)[17–22] or organic/inorganic hybrid materials[23,24], which contain noble or toxic metals, and have limitations in structural and functional tunability, as well as large-scale production owing to the high costs and harsh fabrication conditions. Compared with their inorganic counterparts, pure organic NLO materials exhibited unique advantages for bioimaging[25], including easy fabrication, wide spectral coverage with tunable absorption and emission spectra, mechanical flexibility, and excellent biocompatibility[26–29]. Motivated by previous works, the DAST materials with SHG and two-photon-excited fluorescence (2PEF) properties, namely, utilize low photon-energy visible/NIR excitation to produce high photon-energy emissions[3,30], showcased great potential to be used as fluorescence probes for realizing real-time bioimaging with high spatial resolution. To the best of our knowledge, unfortunately, achieving water/heat/light-stable organic NLO supramolecular materials with high photoluminescence quantum yields (PLQYs) and persistent NLO activity is still very challenging and intractable, not to mention the real application of DAST materials in high-resolution bioimaging[31].

In this work, we successfully fabricate the flexible, large-area (ca. up to 375 cm²) organic NLO membranes composed of DAST@hydroxypropyl-beta-cyclodextrins (HPβCD) host-guest supramolecular complex with intense fluorescence via one-step electrospinning in a high-throughput manner (i.e., at a production rate of ~750 cm² h⁻¹). The crosslinked HPβCD matrix can spatially suppress the aggregation of DAST molecules, which effectively restrict the ACQ effect and isomerization of DAST molecules, inhibit the excited-state rotation from locally excited (LE) state to twisted intramolecular charge transfer (TICT) state, and largely restrain the nonradiative decay caused by TICT state and vibrational relaxation. As a result of spatial confinement, geometric restriction and intra-/intermolecular packing modulation, the DAST@HPβCD membranes show an ~81-fold enhancement in PLQY from 0.9% to 73.5%. Besides, the DAST@HPβCD membranes are not easily decomposed and deactivated even being immersed in water for >4000 hours, which still preserve their NLO properties owing to enduringly fixed non-centrosymmetric alignment of DAST molecular structure. The DAST@HPβCD membranes exhibit exceptional thermotolerance up to 300 °C, and the embedded light-emitting diode (LED) can continuously emit orange light for >20 hours under 10 W input power. The DAST@HPβCD membranes are stable under ultraviolet (UV) light (a power density of 120 mW/cm²) or femtosecond laser (a wavelength of 1000 nm, 140 fs/pulse, 80 MHz repetition), which can preserve 50% of initial PL intensity after continuous UV light irradiation for 974 hours. The DAST@HPβCD membranes emit broadband 2PEF (ca. 520−650 nm) with tunable excitation wavelength from 770 nm to 1000 nm and emit 3PEF at 1590 nm excitation. These excellent polar solvent/thermal/photostability, as well as MPA and upconversion properties make it possible for DAST@HPβCD membranes to achieve bioimaging with a wide optical window, especially under the excitation of the NIR light regions. We successfully apply the biocompatible DAST@HPβCD membranes as the biological stains for two-photon, non-invasive, real-time, and in vivo bioimaging to observe the microbial activities and behaviors of live Escherichia coli (E. coli).

## Results

### Large-scale fabrication of DAST@HPβCD membranes and chemical interactions between DAST and HPβCD

The electrospinning ink was prepared in N₂-filled glovebox (with H₂O < 0.01 ppm and O₂ < 0.01 ppm) by simply mixing HPβCD host molecules, DAST guest molecules, and 1,2,3,4-butane tetracarboxylic acid (BTCA) crosslinking agents together according to weight ratios as described in Methods section (Supplementary Fig. 1), which was then directly used to fabricate large-area, DAST@HPβCD fibers in a roll-to-roll manner with a high production rate of ~750 cm² h⁻¹ (see details in the Methods section). The one-step, single-nozzle electrospinning process was conducted, which was able to produce a 25 cm × 15 cm DAST@HPβCD fibers within 30 min, showing uniform orange light emission over the whole display area under UV light irradiation (Fig. 1a and Supplementary Fig. 2). It is worth pointing out that the electrospinning process lasting for 3 min could also yield a large-area DAST@HPβCD fibers with observable fluorescence (Supplementary Movie 1). Note, as compared to solution growth of NLO single crystals, the electrospinning has a close to 100% high material usage rate which only requires much less precursor ink, and can largely simplify the fabrication process, thus showcasing the unique advantages of realizing high-throughput mass production of NLO fabrics in a low-cost manner. The cost of DAST-based fabrics is estimated to be 2.18 $/m² (Supplementary Table 1). The HPβCD molecule has distinct structural features in terms of its hydrophobic inner cavity (ca. 0.78 nm in diameter) with numerous hydroxyl groups embedded in both its inner and outer surface[32,33]. In this case, the inner cavities of HPβCD could sequester the DAST molecules (ca. 0.58 nm in molecular width) to form DAST@HPβCD host-guest complexes, which were further crosslinked via the esterification reaction between hydroxyl groups of HPβCD and carboxyl moieties of BTCA to form a 3D crosslinking supramolecular framework (Fig. 1a). This, to a larger extent, reinforced the geometric restriction induced by HPβCD scaffolds, and minimizes the intramolecular motion of DAST molecules via modulating the intermolecular packing states. The cross-linking between HPβCD and BTCA can be evidenced by the occurrence of characteristic Fourier-transform infrared (FTIR) peaks at 1725 cm⁻¹ and 1692 cm⁻¹ corresponding to the C=O stretching of carboxylic and ester functional groups in HPβCD@BTCA sample, accompanied by considerably decreased intensity and widened signal of -OH stretching, relative to pristine HPβCD or BTCA sample[34] (Supplementary Fig. 3a, b). We also investigated the reactions between the functional groups of HPβCD and DAST via FTIR study. The broad vibrational bands in the range of 3000−3500 cm⁻¹ can be assigned to O−H stretching vibrations (3391 cm⁻¹) of HPβCD and N−H stretching vibrations (3431 cm⁻¹) of DAST. The vibration bands at 1414 cm⁻¹, 1636 cm⁻¹ and 1652 cm⁻¹ can be ascribed to C−N stretching vibrations and N-H bending vibrations of DAST, as well as O−H bending vibrations of HPβCD, respectively (Supplementary Fig. 3c, d). After forming DAST@HPβCD complex, one could observe the intensities of both the O−H/N−H stretching vibrations around ~3400 cm⁻¹ and the O−H/N−H bending vibrations ~1640 cm⁻¹ increased (Supplementary Fig. 3c), and the sharp decrease of C−N stretching vibrations (Supplementary Fig. 3d). These peak intensity changes suggested that there may be strong hydrogen bonding interactions between DAST and HPβCD, which is advantageous for suppressing the trans−cis isomerization of DAST and thus enhancing its PLQY. We further investigated the host-guest interactions and assembly mode between DAST and HPβCD. The conformation of DAST@HPβCD in an aqueous solution was studied by circular dichroism spectroscopy (Supplementary Fig. 4). For the pure DAST or HPβCD molecule, there is no Cotton effect because DAST easily dissolves in water and HPβCD is not featuring chiral property (Supplementary Fig. 4a). In contrast, in the case of DAST@HPβCD complex with DAST included by HPβCD, the induced circular dichroism (ICD) signal appeared (Supplementary Fig. 4b). There is a strong positive Cotton effect, which corresponded to

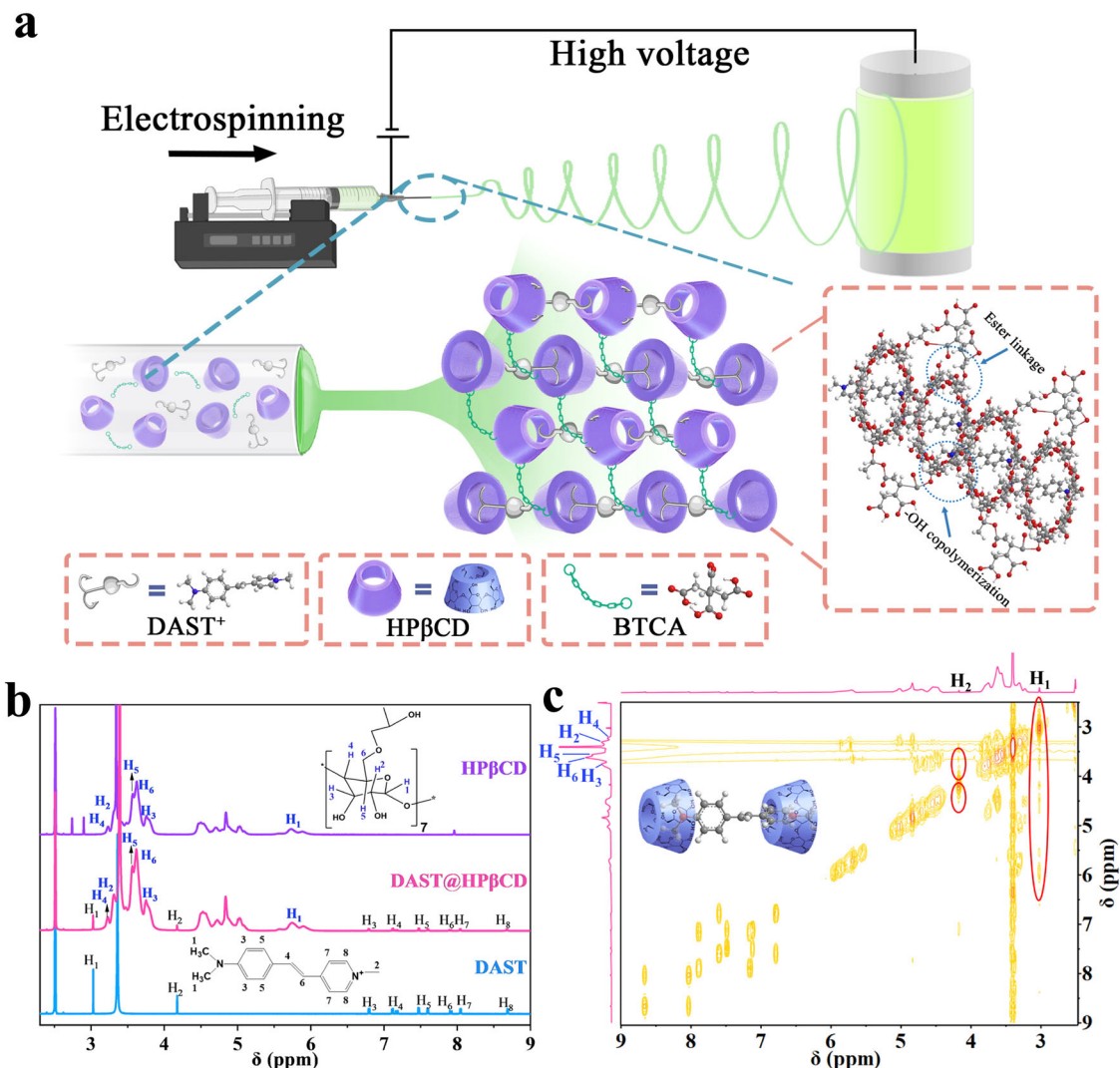

**Fig. 1 | Schematic illustration of the fabrication process of DAST@HPβCD fibrous membranes and NMR characterization. a** Schematic illustration of the electrospinning process for fabricating the DAST@HPβCD fibers. **b** The ¹H NMR spectra (600 MHz, D₆MSO) of DAST, HPβCD, and DAST@HPβCD. **c** The COSY NMR of DAST@HPβCD. The inset showed the insertion mode of DAST molecule in HPβCD, and the tosylate anion has been omitted for better clarity.

the $^1L_a$ band at 220 nm ($\Delta\varepsilon = 0.65$ dm³ mol⁻¹ cm⁻¹) and could be assigned to the pyridine chromophore[35]. As for the negative Cotton effect related to the $^1L_b$ band at -341 nm ($\Delta\varepsilon = -0.56$ dm³ mol⁻¹ cm⁻¹), it could be ascribed to the spectral transition term of benzene ring region. According to Kajtr's sector rule on circular dichroism of cyclodextrin inclusion complex[36,37], the specific orientation of the transition dipole moment of the guest chromophore with respect to the dipole moment of the cyclodextrin host would result in specific sign of ICD signal, either positive or negative. When the guest chromophore was located inside the cavity of cyclodextrin host, a positive ICD signal would be obtained when the electronic transitions of chromophore were parallel to the cyclodextrin axis, whereas a negative ICD signal would be observed under the circumstances of those perpendicular transitions[35,38]. Obviously, the HPβCD molecules provide different interaction microenvironments for the pyridine group and benzene ring groups. We speculated that the positively charged pyridine group of DAST parallelly inserted into HPβCD cavity and the aromatic ring and dimethylaminostyryl of DAST are perpendicular to HPβCD cavity (Supplementary Fig. 5). Such non-planar molecular packing and twisting is able to increase the steric hindrance, and thus effectively suppressing the self-aggregation of DAST molecules. The ¹H nuclear magnetic resonance (¹H NMR) spectra of DAST, HPβCD, and DAST@HPβCD further verified the

specific inclusion mode of HPβCD matrix on DAST molecule. After forming the DAST@HPβCD complex, the resonances of $H_1$, $H_2$, $H_5$, $H_6$, and $H_7$ protons in DAST molecule showed highfield shift, while the resonances of $H_3$, $H_4$, and $H_8$ protons in DAST molecule experienced downfield shifts (Fig. 1b and Supplementary Table 2). This can be explained by the anisotropic effect of aromatic rings and the C=C bonds in DAST molecule during NMR characterization, in which the different protons residing on different spatial locations would encounter either shielding or deshielding effect[39], thus resulting in the chemical shifts of protons to different directions. The different affinity between HPβCD cavity and different functional groups (i.e., the hydrophobic dimethylamino group and hydrophilic pyridine ring) of the DAST molecule may also contribute to the abovementioned anisotropy effect. As for the resonances of $H_2$, $H_3$, $H_4$, $H_5$ and $H_6$ protons residing on the cavity of the HPβCD molecule, they all shifted to highfield directions after DAST insertion (Supplementary Table 3). Overall, the shifts in the ¹H NMR spectra of both DAST and HPβCD after forming DAST@HPβCD composites were related to the hydrogen bonding interactions and electrostatic interactions between these molecules[40]. In addition, the correlation spectroscopy (COSY) ¹H NMR was measured to gain more insight into the geometric structure of the DAST@HPβCD complex. Specifically, strongly coupled signals between both $H_1$ and $H_2$ protons

of DAST with the protons in HPβCD further confirmed the insertion of DAST molecule into the cavity of HPβCD host. Overall, combining the ICD signal, ¹H NMR, and COSY NMR results, we confirmed that the DAST molecules readily reside in the HPβCD cavities to form the DAST@HPβCD supramolecular host-guest inclusion complexes. Supplementary Fig. 6 displayed the X-ray diffraction (XRD) patterns of the pure DAST crystals, HPβCD powders, and DAST@HPβCD fibers, which further verified the successful inclusion of DAST by HPβCD, as evidenced by the disappearance of DAST characteristic peaks and slightly broadening signals in DAST@HPβCD sample.

### Morphological features, energy level, and luminescent property of DAST@HPβCD

Scanning electron microscopy (SEM) images and fluorescence confocal microscopy images visualized the well-interconnected fibrous morphology of DAST@HPβCD membrane, which was woven by individual fibers (~1 μm in diameter) with bright fluorescence (Fig. 2a, b). Interestingly, upon exposure of DAST@HPβCD fibers in a high relative humidity (RH) environment (~70–85%) for 10 mins, it turned into a transparent thin film with dense morphology, which is possibly due to the hydrogen bonding interactions and copolymerization process

triggered in the presence of H₂O (Fig. 2c, d). In this case, the H₂O could not only interact with HPβCD oligomers assembled on the surface of fibers via hydrogen bonding, but also facilitate the self-assembly and/or aggregation of numerous HPβCD molecules (Supplementary Fig. 7)[41], which ultimately promoted copolymerization/interaction among adjacent fibers (i.e., hydrogen bonding between -OH groups of HPβCD molecules). As a result, all of the pores and voids within the interconnected network were filled out and the pristine porous membrane was densified to form a compact and transparent thin film (Supplementary Fig. 8). Notably, the post-annealing treatment of DAST@HPβCD fibrous membrane will block the copolymerization process, and in this case, the individual fibers will not crosslink and merge together to form the compact thin film (Supplementary Fig. 9). The large-area, transparent DAST@HPβCD thin film can also be electrospun on polyimide (PI) substrate (Fig. 2e), which showed a smooth surface with a very small root-mean-square (RMS) roughness of 6.36 nm (Supplementary Fig. 10), which is beneficial to minimize the optical scattering loss and effectively reflect the desired orange emission with great uniformity. The transmission electron microscope (TEM) characterizations revealed the morphology of nano-sized, crystalline DAST@HPβCD composited crystals (i.e., ~12 nm

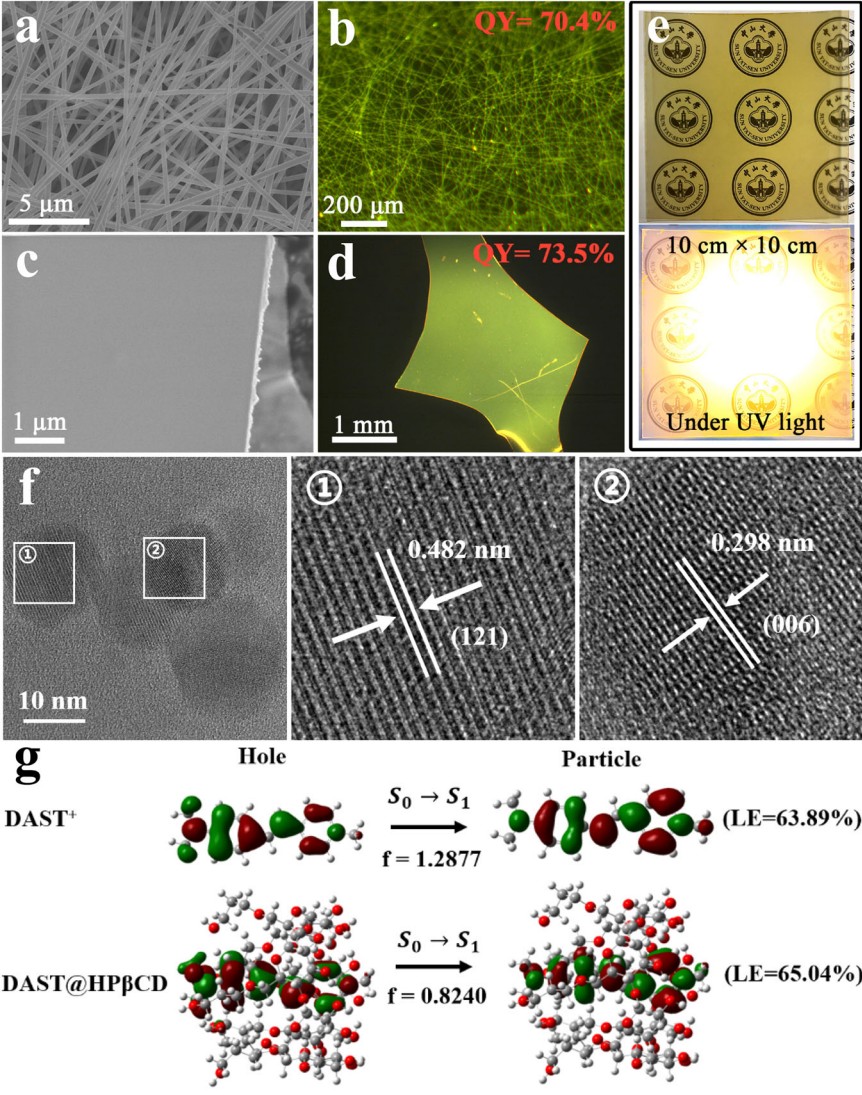

**Fig. 2 | The morphological characterization and NTO simulation. a** SEM image and **b** fluorescence image of DAST@HPβCD fibers. **c** SEM image and **d** fluorescence image of DAST@HPβCD thin film after humidity-triggered copolymerization. **e** The photograph of a large-area (10 cm × 10 cm), transparent DAST@HPβCD thin film on a flexible PI substrate with and without UV light irradiation. **f** TEM image of crystalline DAST@HPβCD composited nanocrystals with their corresponding lattice fringes. **g** NTO simulation of DAST and DAST@HPβCD. Note, the calculation is based on the model of using a single molecule.

in diameter). We identified the well-resolved lattice fringes of 0.482 nm and 0.298 nm could be indexed to the (121) and (006) crystal planes of DAST materials (Fig. 2f and Supplementary Table 4). We found that the lattice fringe of (121) plane of DAST@HPβCD composited nanocrystals was slightly larger than those of previously reported DAST nanocrystals without any chemical modification (i.e., a lattice fringe of 0.445 nm for (121) plane)[10], which can be attributed to the strong host-guest inclusion interaction between DAST and HPβCD. These results indicated the pure organic NLO DAST-based materials and membranes can be facilely prepared via combining host–guest chemistry and electrospinning technology.

DAST, a kind of stilbazolium derivative, is a typical ICT molecule containing a D-π-A-type structure, which normally exhibited an extraordinarily wide luminescence band that is identified as dual luminescence caused by LE and TICT state-stimulated emissions[42]. In general, the LE state emits short-wavelength photoluminescence (PL) and high PLQY, while the TICT state exhibits long-wavelength PL emission and low PLQY due to its quick nonradiative decay to the ground state. Therefore, to improve the fluorescence properties, it is desirable to ensure that the luminescent molecules were mainly excited under LE regime. To further gain insight into the frontier molecular orbital (FMO) distributions and electronic structures of the DAST and DAST@HPβCD, the natural transition orbitals (NTO) simulation was performed using the Gaussian 09 package at the M06-2X/6-311+g (d,p) (see the calculation details in the Methods section)[43–46]. It shows that the hole and particle of $S_1$ state are dispersed on the DAST molecule (Fig. 2g), implying that, ideally, the DAST molecule could achieve LE state-dominant characteristic. For instance, this could possibly happen by dissolving the DAST materials in some kinds of solvents with specific polarity, as long as the molecular isomerization and transition to TICT state could be inhibited in the liquid state (Supplementary Fig. 11). In reality, the DAST material is not efficiently luminescent and loses its NLO properties owing to its vulnerability in ambient air and/or polar solvent environment. For DAST@HPβCD complex, the proportion of LE state increased and the oscillator strength ($f$) severely decreased from 1.2877 to 0.8240, which could be attributed to the increased energy gap with HPβCD inclusion[47]. Note, the above NTO simulation is based on the model of using a single molecule, while in a real case, whether each one has LE state-dominant characteristics should also consider its physical form (solid or liquid), the solvent effect, the specific molecular compositions, and other more complicated factors. The calculation data is consistent with the experimental result of cyclic voltammetry (CV) measurement, which showed the increased optical energy band from 1.61 eV for DAST crystal to 1.97 eV for DAST@HPβCD fiber, and to 2.15 eV for DAST@HPβCD thin film (Table 1 and Supplementary Fig. 12). We attributed the widened optical bandgap to the nanoconfinement of DAST materials by HPβCD inclusion, which resulted in blue-shift of both the absorption and PL spectra (Supplementary Fig. 13). Specifically, the DAST crystal exhibited a large spectral overlap between absorption and luminescence (an overlap area of ~33.22%). Upon the introduction of large amounts of HPβCD molecules, the absorption spectra underwent a blue-shift along with the enhancement of absorption intensity, and a reduced spectral overlap (an overlap area of ~25.10%) was observed, which is beneficial to mitigate the self-absorption effect.

Upon photoexcitation, the excited-state population in DAST crystals is dominated by LE state, which is in favor of electron transfer from the electron-donating moiety to the electron-accepting pyridinium unit, accompanied by TICT state featuring fast nonradiative decay owing to the single/double bond twisting. Nonetheless, DAST crystal still showed ay low PLQY of 0.9%[8] (Table 1) owing to the ACQ effect induced by the dense packing of molecules in the crystal. It is widely demonstrated that immobilizing ICT molecules in the porous framework can mitigate this issue[42,48]. In the case of the DAST@HPβCD membranes, the HPβCD could spatially separate the DAST molecules

and thus minimize the ACQ effect. More importantly, the supramolecular host-guest inclusion complex with abundant intermolecular interactions (i.e., electrostatic interactions, hydrogen bond interactions, etc)[36] can effectively restrict the twisting of DAST molecules, optimize the population distribution between the LE and TICT states, and reduce the nonradiative decay of TICT state (Supplementary Fig. 14). All these characteristics should help to improve the luminescence performances and optical properties. As a result, the DAST@HPβCD fibers significantly amplified the PLQY by ~78-fold, reaching a high value of 70.4% (Fig. 2b and Supplementary Fig. 15). We attributed the remarkable fluorescence enhancement to the HPβCD introduction and inclusion, as is evidenced by the gradual amplification of PL intensity when the molar ratio of host molecules and guest molecules was increased from 0:1 to 12:1 (Supplementary Fig. 16). Subsequent densification process further boosted the PLQY to a benchmark value of 73.5% for the DAST@HPβCD thin film (Fig. 2d, Supplementary Fig. 17 and Table 1). The different optical properties (i.e., absorption and PL peaks, $E$g, PLQY, etc.) for the DAST@HPβCD fiber and DAST@HPβCD thin film can be attributed to their distinct differences in terms of hydration degree[49] and morphologies. Accordingly, we also summarized the differences in 1PEF, 2PEF, and 3PEF properties between DAST and DAST@HPβCD, as can be seen in Supplementary Table 5.

## Luminescent mechanism

The luminescent performance and mechanism of DAST and DAST@HPβCD samples can be understood via temperature-dependent PL characterizations. Under 420 nm light excitation, DAST crystals exhibited dual PL emission at ~615 nm and ~746 nm (Table 1), which corresponded to the luminescence from LE and TICT state[31,50], respectively (Supplementary Fig. 13 and Fig. 3a). In sharp contrast, the DAST@HPβCD fibers only emitted single orange fluorescence at ~583 nm, and consistently exhibited stronger PL intensity when increasing the temperature from 150 K to 450 K (Fig. 3b), indicating the HPβCD inclusion effectively restricted the intermolecular interaction, trans–cis isomerization and forbids excimer formation of DAST molecules[8,42]. Compared with the DAST crystal, the DAST@HPβCD fibers exhibited >~91-fold enhancement of radiative recombination rate ($K_r$) measured at 300 K (Supplementary Table 6). Additionally, the temperature-dependent PL decay lifetimes of DAST and DAST@HPβCD fibers were characterized. In particular, the PL lifetime of DAST crystals always exhibited a biexponential decay process and witnessed a rapid decrease when the temperature varied from 150 K to 400 K (Supplementary Fig. 18 and Supplementary Table 7). In contrast, the PL lifetime of DAST@HPβCD fibers just showed a single-exponential decay process with comparatively less variation when the temperature increased from 150 K to 390 K (Supplementary Fig. 19 and Supplementary Table 8). These results indicated that the crosslinked HPβCD host matrix can localize the LE state and completely inhibit TICT state of the DAST compounds, and also suggested good thermal stability and persistent PL property of as-prepared DAST@HPβCD fibers. Such good thermal stability can be further verified by thermogravimetric (TG) and differential scanning calorimeter (DSC) analysis. As expected, the thermal stability of DAST@HPβCD fibers is much better than that of DAST crystal. Specifically, the DAST@HPβCD fibers can tolerate a high temperature of up to 300 °C without obvious decomposition (~2.9% weight loss), while the DAST crystals witnessed ~18.5% weight loss at 300 °C (Supplementary Fig. 20). For the DSC results, there is an endothermic reaction at ~261 °C for both samples, which referred to the melting point of DAST[51]. For the peak at 133.1 °C showed in DAST@HPβCD fibers, it can be possibly assigned to the breakage of hydrogen bonds or dehydration under the temperature range of ~120-150 °C (Supplementary Fig. 21).

The femtosecond transient absorption (fs-TA) spectroscopy characterizations were further carried out to explore the population distribution process between the LE and TICT states at a timescale

**Table 1 | Optical properties and energy band parameters of DAST crystal, DAST@HPβCD fiber, and DAST@HPβCD thin film**

| Materials | $\lambda_{abs}$[a] [nm] | $\lambda_{em}$[b] [nm] | $\tau_1$/ratio [ns] | $\tau_2$/ratio [ns] | $E_g$[c] [eV] | HOMO[d] [eV] | LUMO[e] [eV] | PLQY [%] |
|---|---|---|---|---|---|---|---|---|
| DAST | 578,661 | 615,746 | 2.284/61.49 | 5.529/38.51 | 1.61 | −5.21 | −3.60 | 0.9 |
| DAST@HPβCD fiber | 457 | 583 | 2.999/100 | 0/0 | 1.97 | −5.56 | −3.59 | 70.4 |
| DAST@HPβCD thin film | 466 | 575 | 2.878/100 | 0/0 | 2.15 | −5.33 | −3.18 | 73.5 |

[a] UV-vis absorption peak and [b] PL peak derived from DAST crystals, DAST@HPβCD fibers, and DAST@HPβCD thin films measured at room temperature; $\tau_1$/ratio and $\tau_2$/ratio are double/single-exponential PL lifetime and corresponding proportion, respectively; [c] Optical energy gap ($E_g$) determined from the UV-Vis absorption spectra (Supplementary Fig. 13); [d and e] Calculated HOMO and LUMO positions from cyclic voltammetry measurement (Supplementary Fig. 12).

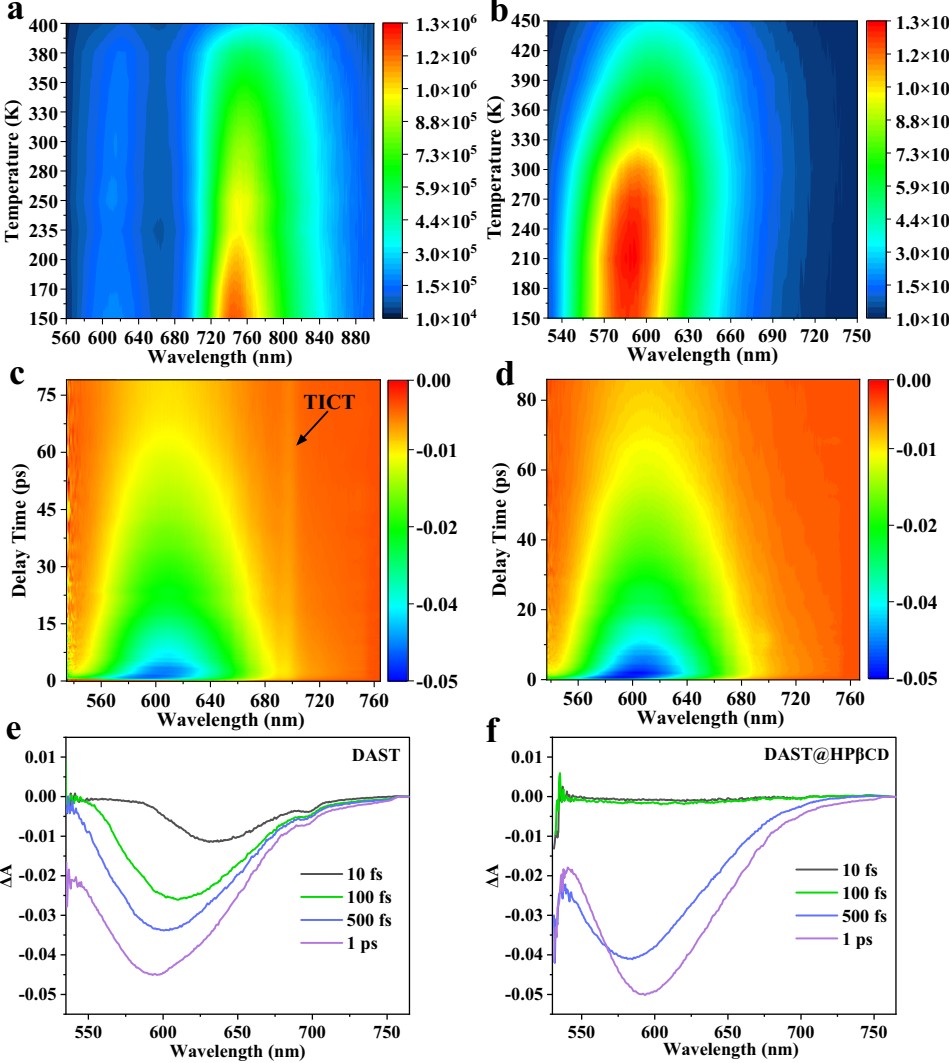

**Fig. 3 | Characterizations of temperature-dependent PL intensity and TA spectra of DAST and DAST@HPβCD samples.** Temperature-dependent PL intensity mappings of **a** DAST crystals and **b** DAST@HPβCD fibers. The contour plots and delay time-dependent TA spectra of **c**, **e** DAST and **d**, **f** DAST@HPβCD samples.

from fs to picosecond (ps) for the DAST molecules with or without HPβCD inclusion. For TA measurement, the DAST crystals and DAST@HPβCD fibers were both dissolved in distilled water (high solution polarity environment). Upon 350 nm fs-laser irradiation, the contour plots of delay time-dependent TA spectra (referred to the change of absorption intensity (ΔA) as a function of delay time from 0 to 75 ps) of both DAST and DAST@HPβCD samples exhibited broad ground-state bleaching (GSB) peaks at ~600 nm (Fig. 3c, d). Specifically, one could notice a narrow, but traceable GSB peak at ~700 nm for DAST sample (Fig. 3c), which is related to the presence of TICT state population. The formation process of the TICT state can be analyzed

by monitoring the photo-induced ΔA[52,53]. For both two samples, the ΔA of the GSB signal gradually increased with prolonged delay time (Fig. 3e, f). For the DAST sample, there are two observable peaks corresponding to the abovementioned LE and TICT state. It is worth noting that the TICT state in DAST sample is easily excited within tens of femtoseconds, especially under a high solution polarity environment, and the DAST molecules could freely relax between the LE state and the TICT state owing to the small energy barrier[54]. Increasing the delay time from 10 fs to 1 ps would excite more TICT state and facilitate its transition to LE state, which resulted in energy redistribution within different excited states and led to bandgap renormalization of DAST

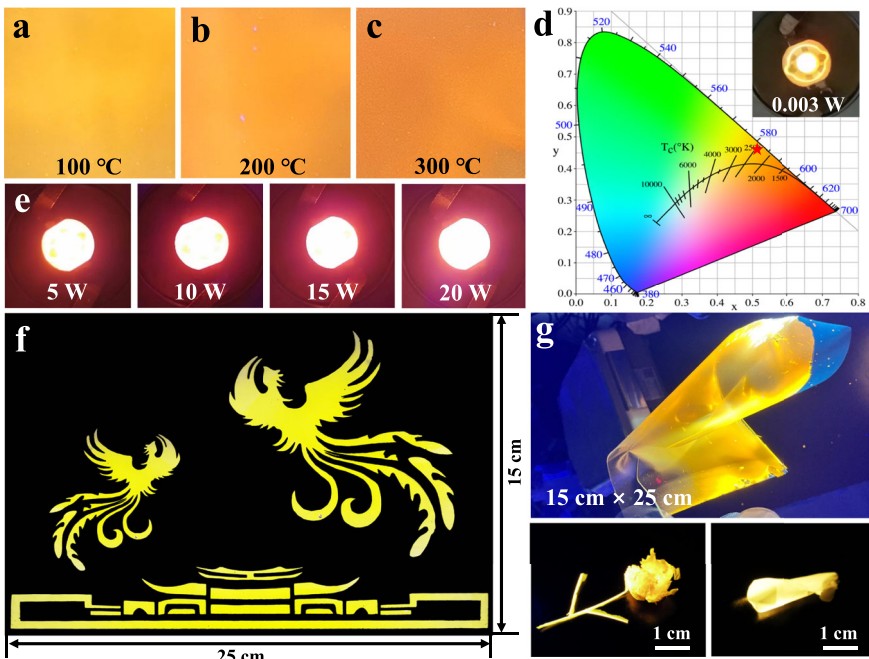

**Fig. 4 | Luminescent stability and applications.** The fluorescence images of DAST@HPβCD thin film (5 cm × 5 cm) heated at **a** 100 °C, **b** 200 °C, and **c** 300 °C. **d** The 1931CIE chromaticity coordinates of orange LED driven by 0.003 W. **e** An orange LED device driven by 5 W, 10 W, 15 W, and 20 W, respectively. **f** Demonstration of large-area patterned displays fabricated with DAST@HPβCD materials. **g** Demonstration of stereo handcrafts made of DAST@HPβCD thin films.

molecules, thus significantly blue-shifting the absorption peaks[55] (Fig. 3e). In contrast, in the case of DAST@HPβCD sample, there is only one peak attributed to LE state, regardless of extending the delay time (Fig. 3f). As mentioned above, the HPβCD inclusion is able to create a high energy barrier that confines the transition from the LE state to TICT state. In the case of DAST@HPβCD complex featuring robust intermolecular host-guest interaction that is easily vibrationally excited[56], prolonging the delay time (i.e., from 500 fs to 1 ps) could cause energy accumulation in LE state, thus red-shifting the absorption peaks and enhancing the ΔA. These results again verified the relatively low polarity of the HPβCD cavity could effectively inhibit the non-radiative transition of TICT state and promote the localization of LE state, thus remarkably enhancing the PLQYs[11,42].

## Luminescent stability and applications

Distinguished from the previously reported DAST or their analogs-based supramolecular complex microcrystals with a host-guest stoichiometric ratio of 1:1 or 2:1 and the highest PLQYs of 10.2%[42], our demonstrated DAST@HPβCD fibers or thin films are composed of an excessive amount of HPβCD host molecules and a minority of DAST guest molecules (i.e., the molar ratio of HPβCD molecules and DAST molecules was approximately 109:1, which is expected to be higher than the real host-guest stoichiometric ratio of forming DAST@HPβCD, since not all the excessive HPβCD molecules were consumed to form DAST@HPβCD and a large portion of the HPβCD molecules were consumed to form the nanofiber skeleton/backbone), which, surprisingly, exhibited high PLQYs over 70%, thus representing a significant advance in this field. In addition to such high PLQYs, our demonstrated luminescent DAST@HPβCD membranes also conferred good ambient and thermal stability, which are crucial to achieving practical lighting applications. Specifically, the DAST@HPβCD fibers preserved 88% of their initial PLQY upon storage in ambient air (i.e., ~20–30 °C and ~50–80% relative humidity, Supplementary Fig. 22) for 470 days. Temperature-dependent luminescent property and fluorescence images of DAST@HPβCD thin film electrospun on PI substrate were investigated, which all showed orange luminescence over the temperature ranges of 100-300 °C (Fig. 4a–c, Supplementary Fig. 23).

Notably, even when the temperature reaches 300 °C, the PLQY of DAST@HPβCD thin film still reached 67.8% (Supplementary Fig. 24), indicating the robust thermotolerance, which outperforms other reported ICT materials[57–59], such as organic conjugated deep-blue nano-emitters like MC8TPA and OEYTPA (thermal tolerance up to ~120 °C)[58], intermolecular coupling of OEYTPA and cmTPA materials (thermal tolerance up to ~160 °C)[59]. Such good ambient and thermal stability is related to the high chemical and thermal stability of cross-linked HPβCD matrix, and its robust supramolecular encapsulation that inhibits the structural deformation of DAST molecules under the high humidity condition, as well as effective inclusion via multiple interactions that suppresses intramolecular thermal motion of DAST molecules upon heating. Specifically, the crosslinked HPβCD matrix effectively inhibited the reactions of DAST with $H_2O$ in several ways. First of all, the HPβCD host can effectively confine the DAST molecules within the hydrophobic inner cavities. Secondly, the HPβCD inclusion can modify the local environment around DAST molecules and impart geometric constraints, which reduced the reactivity of the DAST with $H_2O$. Last but not least, the crosslinked HPβCD matrix that exposed hydrophobic moieties as much as possible could serve as a robust protection barrier to prevent potential water invasion to the encapsulated DAST molecules, thus reducing the likelihood of decomposition or other unwanted reactions. Considering the promise of the abovementioned high thermotolerance, we fabricated the orange LEDs by directly pasting a small piece of DAST@HPβCD thin film on the commercial UV (380 nm) LED chip (Fig. 4d). The orange LED can be lightened up at a low-power of 0.003 W (i.e., a current of 1 mA and a voltage of 3 V was applied, inset in Fig. 4d), and the Commission International L'Eclairage (1931CIE) chromaticity coordinates are (0.520, 0.467), suggesting the capability of serving as low-power-charging LEDs. The orange fluorescence became much stronger when increasing the input power from 5 W to 20 W (Fig. 4e), demonstrating the high-power-driving capability of DAST@HPβCD thin film-based LEDs. It is worth pointing out that our orange LEDs can operate at a high working current of up to 100 mA. In addition, the DAST@HPβCD-based orange LED operating at 10 W still retained very strong luminescence for 20 h. This, again highlights the promise of applying these

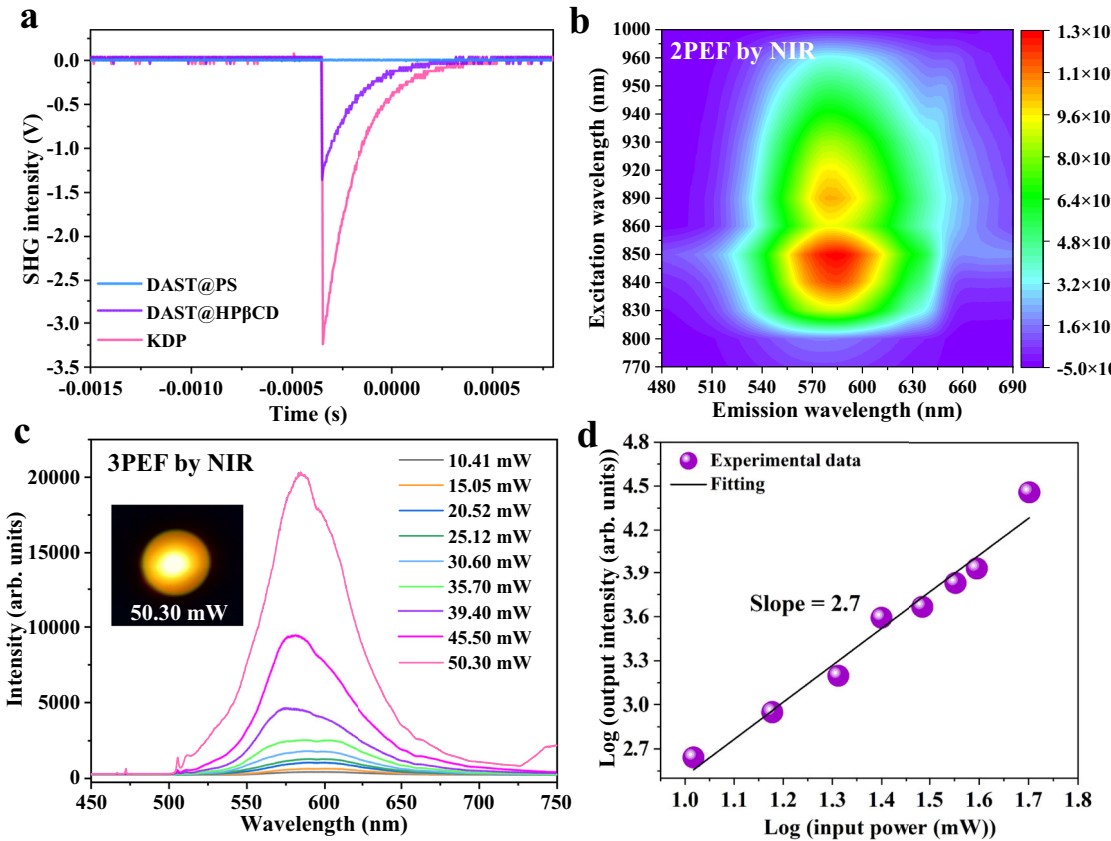

**Fig. 5 | The NLO property of DAST@HPβCD fibers. a** SHG signals of the reference KDP, DAST@HPβCD fibers, and DAST@PS fibrous membrane, respectively. **b** Excitation wavelength-dependent 2PEF of DAST@HPβCD fibers. **c** 3PEF spectra of the DAST@HPβCD fibers measured at different input laser intensities, in which the inset shows the 3PEF image excited by a NIR laser with a power of 50.30 mW. Notably, the diameter of the laser spot is 0.40 mm. **d** The logarithm of the 3PEF intensity is a function of the logarithm of the input laser power.

host-guest supramolecular complexes in sustainable and durable lighting applications with a wide operating current window (1-100 mA) and extended lifespan. On basis of the advantages of the large area, high flexibility, and good transparency, the DAST@HPβCD thin film can be applied in patterned displays and decorating, namely, two phoenixes flying on top of a palace (Fig. 4f). Many stereo handcrafts made of DAST@HPβCD thin films were created by simple rolling, folding and bending, for instance, a golden rose (Fig. 4g). Overall, the large-area, flexible, transparent, DAST@HPβCD thin films are highly desirable and promising for wearable electronics, LEDs, panel displays and photoelectric signal conversion.

## NLO property

We expected fascinating NLO property for DAST@HPβCD host-guest supramolecular complex owing to the enduringly fixed noncentrosymmetric arrangement of DAST molecular structure by crosslinked HPβCD host matrix, regardless of in solid-state or solution-state (i.e., dissolved in water). Raman spectra showed that, in water environment, only when the DAST is included by HPβCD, in which the trans−cis isomerization of stilbazolium is largely suppressed, its characteristic peaks preserve and show up (Supplementary Fig. 25). This result confirmed the crosslinked HPβCD matrix could achieve robust supramolecular encapsulation of DAST molecules, which prevented them from decomposition or forming hydrates even in high humidity conditions. We measured the SHG, 2PEF, and 3PEF performances of DAST@HPβCD fibers. The SHG testing was performed by applying the Kurtz and Perry model (λ = 1064 nm, Nd: YAG pulsed laser)[60]. As shown in Fig. 5a, 0.41 times of SHG signal was detected for the DAST@HPβCD fibers, relative to the standard, commercialized potassium dihydrogen phosphate (KDP). To verify how the HPβCD inclusion played a synergistic role in generating the SHG signal, we fabricated the DAST@polystyrene (PS) fibrous membrane (Supplementary Fig. 26) for comparison. As expected, there is no SHG signal in DAST@PS membrane (Fig. 5a). Such a comparison suggested the uniform molecular orientation of DAST molecules in the host-guest supramolecular complex, while the DAST molecules randomly dispersed and mixed with PS polymer in a disordered manner owing to the lack of robust host-guest interactions. When using an NIR laser as the incident light, efficient 2PEF was detected for the DAST@HPβCD fibers. In general, the 2PEF is excited by a two-photon absorption (2PA) process with the simultaneous absorption of two photons via a virtual state, which is more complicated than the typical one-photon absorption (1PA) process[61]. Notably, the 2PEF excitation wavelength of DAST@HPβCD fibers can be extended from 770 to 1000 nm (Fig. 5b), thus showing good adaptability of the excitation-energy range. Furthermore, the two-photon action cross-section (TPACS)[62] of DAST@HPβCD fibers showed obvious fluctuation with the change of excitation wavelength (Supplementary Fig. 27). When being excited by 820 nm laser, TPACS reached a maximum value of 186.1 G.M. These results illustrated that the HPβCD matrix inclusion not only enabled to achieve a high PLQY via one-photon-excited emission, but also realized a striking 2PEF at a broadband excitation window covering the whole short-wavelength NIR regions. In addition, a three-photon absorption (3PA) process occurs when the DAST@HPβCD membrane is excited by 1590 nm laser. The setup was illustrated in Supplementary Fig. 28, in which a fs pulsed laser (1590 nm, a repetition rate of 1 kHz, and a pulse duration of 40−60 fs) was utilized as the pumping source. The dependence of the 3PEF intensity (I) on the laser power intensity was displayed

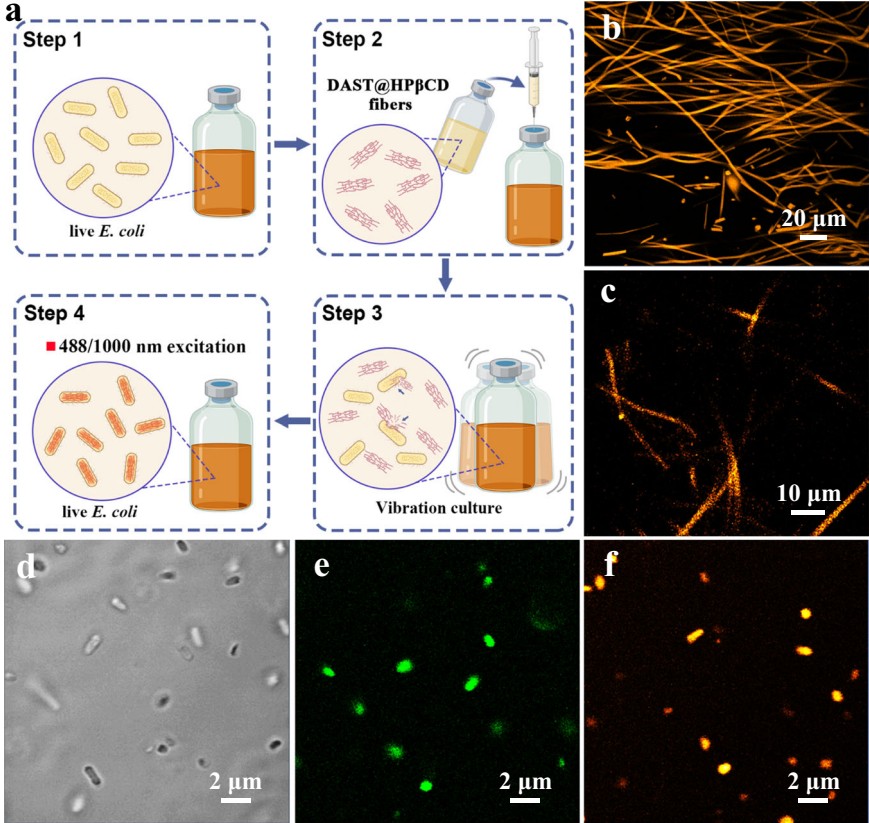

**Fig. 6 | Demonstration of in vivo bioimaging of *E. Coli* via DAST@HPβCD fibers labeling and corresponding CLSM images. a** The schematic illustration of in vivo bioimaging of *E. coli* by employing DAST@HPβCD fibers. The CLSM images of *E. coli* incubated with the DAST@HPβCD fibers for **b** 8 h and **c** 24 h under 1000 nm excitation, and for 96 h under **d** bright-field mode, **e** 488 nm excitation, and **f** 1000 nm excitation. Note, the fluorescence images captured at 488 nm or 1000 nm excitation are pseudo-colored for better distinction.

(Fig. 5c), and the corresponding 3PEF images were shown in Supplementary Fig. 29. As the pump power increased, the emission light became more bright, as is evidenced by enhanced 3PEF intensity (Fig. 5c and Supplementary Fig. 29). A linear log($I$)-log($P$) relationship can be established between the 3PEF intensity ($I$) and laser power ($P$), and the slope of the fitting line is 2.7 (Fig. 5d), which is close to 3, thus confirming that the upconversion-emitted luminescence is indeed induced by a nonlinear 3PA process.

**Two-photon bioimaging**

The DAST@HPβCD fibers displayed excellent NLO properties with enhanced SHG, 2PEF, and 3PEF performances, making them potential candidates for bioimaging applications. The key to making it feasible is to maintain these desirable NLO properties of DAST@HPβCD in an aqueous solution. As mentioned above, it is well-known that the stereotyped DAST materials/crystals would lose most of their NLO properties in a polar solvent, as is characterized by very weak emission, owing to the structural deformation and/or rapid twist relaxation from LE to TICT states[8]. Surprisingly, our fabricated DAST@HPβCD fibers still showed excellent luminescent stability even after being soaked in water for 4000 hours, continuously emitted orange fluorescence and persevered ~60% of its initial PL intensity (Supplementary Fig. 30). Under such circumstances, the SHG signal was still clearly observed (Supplementary Fig. 31). These results indicate that the sensitivity of DAST material to polar solvent has been effectively mitigated. This can be attributed to the robust encapsulation and protection of DAST molecules by a large portion of hydrophobic HPβCD cavities and the abovementioned enduringly fixed non-centrosymmetric arrangement of DAST molecular structure. Though promising, a decrease in the SHG signal of DAST@HPβCD was still observed upon water immersion for

quite a long time (i.e., 4000 hours, Supplementary Fig. 31). This can be attributed to the breakage of the crosslinking network, possibly owing to the undesirable ester hydrolysis. In this case, the inner DAST materials were readily accessible by the $H_2O$ molecules, leading to the decomposition of DAST and/or the formation of DAST hydrate, resulting in a gradual loss of SHG signals with continuously extended water immersion time. However, in practical utilization of the SHG property for upconversion application, the solid-state DAST@HPβCD sample does not need to be directly immersed in water, as it is believed to be stable enough to maintain the SHG signals and exhibit a largely extended lifespan. Another promising aspect of the DAST@HPβCD complex is its 2PA attribute, which enables two-photon fluorescence imaging excited by the NIR pulsed laser. This is advantageous in terms of achieving lower photodamage, deeper penetration, and higher spatial resolution for bioimaging, especially at the cellular level, in tissues and living bodies. In this regard, one should also consider photostability, another important criterion that determines the potential of applying the DAST-based materials as luminescent probes for practical use. We investigated the photostability of DAST@HPβCD fibers under continuous UV (a power density of 120 mW cm⁻²) and fs-laser irradiation. Encouragingly, the DAST@HPβCD fibers exhibited good photostability under continuous UV irradiation for 200 h (Supplementary Fig. 32a), and the extrapolated time drop down to 50% PL (TPL₅₀) is 974 h (Supplementary Fig. 32b). Furthermore, the DAST@HPβCD fibers soaked in water also preserved their luminescent property (i.e., brightness) and 2PEF when continuously irradiated by 488 nm and 1000 nm laser for 30 mins (Supplementary Fig. 33).

As a proof-of-concept, we applied the DAST@HPβCD fibrous membrane to the real-time imaging of live *E. coli* (see details in the Methods section). The process of the one/two-photon in vivo

bioimaging test was schematically illustrated in Fig. 6a. Specifically, 10 ml of alive *E. coli* was incubated with 5 mg of DAST@HPβCD fibers. The *E. coli* was illuminated after digesting the DAST@HPβCD fibers. The confocal laser scanning microscopy (CLSM) images (collected at bright-field (black and white color), 1PEF at 488 nm (pseudo-green color), and 2PEF at 1000 nm (pseudo-orange color) of the *E. coli* after incubation with DAST@HPβCD fibers for different durations are shown in Supplementary Fig. 34. Note that pseudo-colored staining was employed to better distinguish the 1PEF and 2PEF fluorescence images, though generally the real colors of 1PEF and 2PEF are very close or similar[8]. In Stage I (0-8 h), the fibrous morphology of DAST@HPβCD was maintained, rather than being consumed by *E. coli*. We speculated that *E. coli* preferred to utilize the more accessible carbon sources from the substrates (e.g., tryptone and yeast extract) to sustain themselves instead of fluorescent fibers at the beginning of the incubation process (Supplementary Fig. 34a–c and Fig. 6b). In Stage II (8-24 h), the fibers were gradually bitten and consumed by *E. coli*, as evidenced by the gradual disappearance of fluorescent fiber fragments (Supplementary Fig. 34d–f and Fig. 6c). Interestingly, unlike the other biological stains that work through cellular injection or surface binding (i.e., physical absorption or covalent bonding), an allodial uptake of DAST@HPβCD fibers by *E. coli* was observed, suggesting the great potential for non-invasive bioimaging. In Stage III (24-96 h), the DAST@HPβCD fibers were entirely phagocytosed by *E. coli* (Supplementary Fig. 34g–i and Fig. 6d–f), targeting the cytoplasms of the *E. coli* cells with bright-filed (Supplementary Fig. 35), 1PEF (Supplementary Fig. 36) and 2PEF (Supplementary Fig. 37) emission. In comparison, the DAST@HPβCD fibers were not decomposed or consumed in the culture solution without *E. coli* inoculation and kept emitting bright fluorescence after 96 h (Supplementary Fig. 38), which again confirmed the material stability in a polar solution and further verified that the consumption of DAST@HPβCD fibers should be ascribed to the bacterial activities.

Given that HPβCD is a kind of cyclic oligosaccharide, a natural product of starch, the outward HPβCD can be directly consumed by *E. coli* as an organic carbon source. Afterward, HPβCD could be completely metabolized by *E. coli*, while the fluorescent DAST@HPβCD would be internalized in the cells and stained them. Supplementary Movie 2–4 showed the in vivo real-time imaging of fluorescence-labeled *E. coli*, and the snapshots were taken at bright-filed, 488 nm excitation, and 1000 nm excitation, respectively (Supplementary Figs. 35–37). It could be noted that more luminescent *E. coli* cells with stronger fluorescence could be observed from 2PEF at 1000 nm than from 1PEF at 488 nm within the same capture area, which can be attributed to the fact that the NIR laser is more harmless and can penetrate deeper to the microbes during imaging observation. As shown in Supplementary Table 9, the state-of-the-art inorganic transition metal chalcogenides quantum dots/nanocrystals, organic fluorescence materials, or organic/inorganic hybrid materials for bioimaging applications have been summarized. Compared with those reported semiconducting quantum dots or pyrrole derivatives-based two-photon bioimaging excited at ~800 nm[18,19,21], our developed DAST@HPβCD-based 2PEF bioimaging can be conducted with more gentle ~1000 nm excitation, which demonstrated great potential to be the next-generation, non-invasive biological imaging materials that ensure much deeper tissue/cell penetration (i.e., >500 μm) and higher spatial resolution[23,63]. As mentioned above, the DAST@HPβCD fibers were highly stable when subjected to NIR excitation, showing a negligible change in emission with laser excitation at ~1000 nm over 30 min (Supplementary Fig. 33), which showcased great potential for long-term tracking of labeled cells. The durability of the fluorescent labeling on *E. coli* cells was evaluated via the fed-batch experiments, where the *E. coli* microbes were continuously fed with the Lysogeny broth (LB) medium stains of 2 mL d$^{-1}$ for 14 days. The labeled *E. coli* can still be clearly traced without observable fluorescence quenching (Supplementary Fig. 39), which suggests good

biocompatibility of DAST@HPβCD fibers. To investigate the possible cytotoxicity of DAST@HPβCD fibers on *E. coli*, we conducted the cell viability assays with increasing concentrations of the luminescent fibers, and evaluated its impact on the growth of *E.coli* (by tracing the change of optical density at a wavelength of 600 nm ($OD_{600}$, the characteristic absorption peak of *E. coli*)) and the fluorescence performance of labeled *E.coli*. Typically, 10 ml of alive *E. coli* was incubated with 0–20 mg of DAST@HPβCD fibers. As a result, there was almost no difference in the growth of *E. coli*, regardless of different concentrations of DAST@HPβCD fibers (Supplementary Fig. 40). The *E. coli* labeled with an increased amount of DAST@HPβCD fibers (i.e., 20 mg fibers) exhibited brighter fluorescence, either being excited at 488 nm or 1000 nm (Supplementary Figs. 41–43). These results suggested the low cytotoxicity of DAST@HPβCD fibers, which are promising to be used as biological fluorescent labeling reagent. Considering an extensive amount of fluorescent *E. coli* cells were found in the medium after a fortnight-fed-batch experiment, the possible conveyance of the fluorescence capacity to the next-generation cells was assumed through fluorescent fiber distribution during mitosis, which could be further studied in the future. Besides, this unique fluorescent labeling method makes it possible to illuminate the bacteria and live-cell, which represents a big step forward for organic NLO materials to be used for in vivo non-invasive bioimaging, and potentially, aiding in surgery and identifying the disease. This biocompatible live-cell fluorescence labeling could also be employed for different research purposes, for instance, to trace the microbial immigration processes, describe the dominant species evolution in microbial consortia, and investigate the microbial invasion mechanisms and patterns. We expected that further behavioral analysis of specific functional microbes in complicated microbial consortia could be guided by our demonstrated 2PEF-assisted bioimaging.

## Discussion

In summary, we demonstrated to fabricate the large-area, flexible, and transparent DAST@HPβCD membranes/thin films with supramolecular host-guest inclusion structure and MPA characteristics via a low-cost, high-throughput electrospinning method. For the DAST@HPβCD complex, the excited-state population was mainly confined in LE states owing to the effective spatial confinement and restricted intermolecular motion of DAST molecules by HPβCD inclusion, while the TICT and other energy loss channels can be effectively suppressed even under harsh conditions. The 1PEF luminescence performance has been improved by 81-fold, and a record high PLQY of 73.5% has been achieved. The DAST@HPβCD thin film is thermotolerant even at 300 °C, enabling high-power-driven (-20 W) orange LEDs with a prolonged operating lifespan. The non-centrosymmetric alignment structure of DAST molecules can be enduringly fixed by crosslinked HPβCD supramolecular network, which endows DAST@HPβCD materials with excellent NLO properties, regardless of in solid-state or solution-state. The DAST@HPβCD composites showed good stability in terms of their luminescent properties when being stored in ambient air, soaked in water, or continuously irradiated by UV/NIR light. The biocompatible DAST@HPβCD fibers were used for in vivo, non-invasive, real-time two-photon bioimaging excited by ~1000 nm NIR laser with lower photodamage, higher penetrability, and signal-to-noise ratio. This work enriches the family of NLO materials, makes it possible to realize low-cost, large-area fabrication for commercialization, and showcases great promise and potential of them in a variety of innovative applications, including lighting, decorating, patterned displaying, anti-counterfeiting, as well as biomedical imaging and theranostics.

## Methods

We confirmed that our research complies with all relevant ethical regulations, and the School of Chemistry, Sun Yat-sen University approved the study protocol.

## Materials

4-N,N-dimethylamino-4´-N´-methyl-stilbazolium tosylate (DAST, 99.5%) crystals were purchased from Sekisui Medical Co. Ltd. Hydroxypropyl-β-cyclodextrin (HPβCD) and 1,2,3,4-butane tetracarboxylic acid (BTCA, 99.8%) were purchased from Shanghai Macklin Biochemical Co., Ltd. N, N´-dimethylformamide (DMF, 99.9%) was purchased from Xi'an Polymer Light Technology Corp. All chemicals were used as received without further purification.

## Fabrication of large-area DAST@HPβCD membranes/thin films (15 cm ×25 cm in size)

Firstly, 10% BTCA (w/w, according to HPβCD) was mixed with 30% HPβCD (w/w, according to DMF) and dissolved in 10 g DMF solution. Then, 0.01 g DAST powder was added to the mixture solution and stirred for 6 hours at 70 °C to obtain the electrospinning ink. The preparation of electrospinning ink was conducted in an $N_2$-filled glovebox (with $H_2O < 0.01$ ppm and $O_2 < 0.01$ ppm). For DAST@HPβCD membrane preparation, 1 g of electrospinning ink was poured into a syringe with a metallic needle (an inner diameter of 0.45 mm). The syringe was positioned horizontally on a syringe pump, and the electrode at the high-voltage power supply was clamped to the metal needle tip and ground to an aluminum collector which was wrapped with aluminum foil. The electrospinning was performed with the following parameters: an applied voltage of 10 kV, a rotation speed of 600 rpm/min, a tip-to-collector distance of 10 cm, and a solution flow rate of 0.5 mL/h. The electrospinning apparatus was enclosed in a Plexiglas box and the electrospinning was carried out at 22–26 °C under 50–70% relative humidity. When placed in a high humidity environment (>~85% RH), the DAST@HPβCD fibrous membrane can quickly crosslink to form a transparent but compact thin film. To further improve the stability of DAST@HPβCD thin film in water, additional thermal annealing at 190 °C for 4 hours was required.

## LEDs fabrication

10 mg of DAST@HPβCD thin film was mixed with epoxy resin glue and pasted onto 380 nm UV-emitting InGaN chip driven by 0.003 W, 5 W, 10 W, 15 W, and 20 W.

## Characterization

The UV-vis absorption (Shimadzu) equipped with an integrating sphere was used to characterize the absorption of DAST crystals and DAST@HPβCD samples. The morphological features of the samples were characterized using a cold-field emission scanning electron microscope (SEM, Regulus 8230), which is equipped with a spatial resolution of 10 nm flat-inserted energy spectrum. The conformation of DAST@HPβCD in aqueous solution was studied by circular dichroism spectroscopy (Applied Photophysics, Chirascan plus). The nuclear magnetic resonance ($^1H$ NMR) and proton/correlation spectroscopy (COEY) characterization were performed on AVANCE NEO 600 M of Bruker, and the deuteroxide ($D_2O$) (Merck) and deuterated dimethyl-sulfoxide (DMSO–$D_6$) (Merck) were used as a reference NMR solvent, respectively. Raman spectra were measured with a Renishaw Raman microscope and spectrometer. The spectral range was adjusted from 100 cm$^{-1}$ to 3100 cm$^{-1}$, and the spectral resolution is 1 cm$^{-1}$. All samples were excited with 785 nm laser beam. Ten cycles of data collection were implemented to reduce the effect of background noise caused by strong fluorescence. The XRD analysis was collected by Rigaku (D-MAX 2200 VPC), the 2θ range was 10 to 45°, with a step size of 0.02° and a step rate of 10 s. Fourier-transform infrared (FTIR) spectra were obtained by using a Frontier FTIR spectrometer. The microstructures of the fibers were examined using a spherical aberration-corrected transmission electron microscope (JEM-ARM200P) operating at an accelerating voltage of 200 kV. The absolute PLQY values were assessed by C 9920-02 absolute PL quantum yield measurement system (Hamamatsu Spectral photometry C9920-02G). An Edinburgh Instruments Ltd

FLS1000 was used to test the steady-state photoluminescence (PL), temperature-dependent PL intensity mapping, and temperature-dependent PL decay lifetime monitoring. The thermal stability of DAST fibrous film was characterized by a differential scanning calorimeter (DSC, Perkin-Elmer Dsc-7) and thermogravimetry (TG, NetzschTG-209) analyzer. The confocal laser scanning microscopy images were taken using Zeiss LSM 880 NLO confocal microscope, in which $\lambda_{two-photon, ex} = 1000$ nm, $\lambda_{em} = 560$–590 nm, 140 fs/pulse, 80 MHz repetition; $\lambda_{one-photon, ex} = 488$ nm, $\lambda_{em} = 550$–600 nm. The CV curves were obtained by virtue of Bio-Logic VMP-300 multichannel electrochemical workstation using the three-electrode cell with a rate of 50 mV s$^{-1}$ in $CH_3CN$ solution (TBAP, 0.1 mol mL$^{-1}$). The DAST@HPβCD electrospinning ink and DAST/N, N-Dimethylformamide (DMF) solution were spin-coated onto the ITO glass substrate as the working electrode. The Ag/AgCl and KCl (sat.) worked as the reference electrode and a platinum disk was used as the auxiliary electrode. It is noted that the HOMO level of external standard ferrocene/ferrocenium (Fc/Fc$^+$) is measured to be −5.10 eV versus Ag/AgCl in $CH_3CN$. The HOMO and LUMO energy levels of samples were calculated by the following equations:

$$HOMO = - [E_{ox\ onset} - E_{ferrocence} + 4.8] \quad (1)$$

$$LUMO = HOMO + E_g \quad (2)$$

$$E_g = 1240/\lambda_{edge} \quad (3)$$

A femtosecond (fs) laser with a wavelength of 1590 nm, a repetition rate of 1 kHz, and pulse widths of 40–60 fs were employed as the light source for 3PEF study. A polarization beam splitter (PBS) was utilized to separate the incident light into two vertically linear polarized light beams. Half-wave plate (HWP) and a quarter-wave plate (QWP) were used to continuously vary the polarization direction of the incident light. The 3PEF signal from the material was collected by an objective, ×50, NA = 0.75 (Zeiss, Epiplan) and detected using a spectrometer with 0.226 nm resolution (Idea optics PG2000 pro). A bandpass filter (PW-532LGP-Y25) was used to remove the excitation laser light. The 3PEF images were taken by the CCD camera (Retiga R6).

## Femtosecond transient absorption (fs-TA) spectroscopy characterizations

The fs-TA measurement was conducted by equipping a regeneratively amplified Ti:Sapphire laser source (Coherent Legend, 800 nm, 150 fs, 5 Jm pulse$^{-1}$, and 1 kHz repetition rate) and Helios (Ultrafast Systems LLC) spectrometers. A 75% portion of the 800 nm output pulse was frequency-doubled in a $BaB_2O_4$ (BBO) crystal, which could generate 400 nm pump light. Meanwhile, the remaining portion of the output was concentrated into a sapphire window to produce a white-light continuum (320-780 nm) probe light. The 350 nm pump beam was generated as part of the 700 nm output pulse from the amplifier, and its power was adjusted by a range of neutral-density filters. The pump beam was focused at the sample with a beam waist of 360 μm; the power intensity was fixed at 14 μJ cm$^{-2}$ in this experiment. With the aid of the mechanical chopper, the pump repetition frequency was synchronized to 500 Hz. The probe and reference beams could be split from the white-light continuum and sent into a fiber optics-coupled multichannel spectrometer by complementary metal-oxide-semiconductor sensors with a frequency of 1 kHz. The samples were prepared by dissolving the DAST@HPβCD composites and DAST crystals in water.

## Two-photon in vivo bioimaging test

*E. coli* (BNCC133264, BeNa Culture Collection) was selected as the typical microbe for bioimaging tests. Luria−Bertani medium was prepared for bacterial growth, containing 10 g/L tryptone, 10 g/L NaCl,

5 g/L yeast extract. After the medium was autoclaved (0.5 MPa, 121 °C, 15 min), its pH was adjusted to 7.4 ± 0.1. The bacterium was first cultivated for 24 h and then subcultured in closed serum bottles in triplicate with an inoculum size of 10%. The bottles were shaken at 100 rpm in an incubator with the temperature controlled at 37 ± 0.5 °C. After 24 h incubation, 5 ± 0.1 mg DAST@HPβCD fibers were added to each bottle (10 mL) to label the *E. coli* cells. The microbial samples (0.2 mL of the supernatant) in each bottle were collected at 8, 24, and 96 h after the addition of fluorescent fibers for evaluating the stain performance using a confocal laser scanning microscopy (Zeiss LSM 880 NLO confocal microscope).

## Computational details

The geometry optimizations of DAST and DAST@HPβCD were carried out by the self-consistent charge density-functional tight-binding (SCC-DFTB) method with a dispersion correction using the DFTB+ program. The smart algorithm[43] with the force convergence tolerance of 0.05 kcal/mol/Å is used and the SCC tolerance is set to be $1.0 \times 10^{-5}$ electrons. Based on the optimized geometries, single point calculation of each complex was carried out at the M06-2X[44]/6-311+g(d,p) level of theory corrected with the Grimme's dispersion (D3)[45] using the Gaussian 09 software package[46]. The natural transition orbital (NTO) distributions and vertical excitation energies were calculated at the b31yp/6-31 g (d) energy level.

## Calculation of two-photon action cross-section (TPACS)

The following formula was used to estimate the TPACS of DAST@HPβCD thin film using Rhoadamin B (RhB$^+$) as the reference,

$$\frac{F_{(reference)}}{F_{(sample)}} = \frac{(\eta\sigma)_{reference} \cdot \rho_{reference} \cdot (I_{00}^2)_{reference}}{(\eta\sigma)_{sample} \cdot \rho_{sample} \cdot (I_{00}^2)_{sample}} \qquad (4)$$

where $F_{(reference)}$ and $F_{sample}$ are the measured 2PEF strengths of RhB$^+$ reference and DAST@HPβCD thin film, respectively. The integrated area below the 2PA excited PL spectra is used for the computation. $\eta$ is the PLQY, $\sigma$ is the two-photon absorption cross-section, $\eta\sigma$ is the action cross-section of the compound, $\rho$ is the sample molar concentration, and $I_{00}$ is the peak intensity of the input laser pulse. Typically, $I = 125.3$ mW (for sample and reference); $\eta_{(reference)} = 0.01$ (RhB$^+$ powder); $\eta_{(sample)} = 0.735$ (DAST@HPβCD); $\rho_{(RhB+)} = 3.74 \times 10^{-5}$ and $\rho_{(sample)} = 2.32 \times 10^{-5}$.

## Statistics and reproducibility

The morphological images of different samples were repeatedly taken more than three times, which all showed similar morphological features of the samples as shown in Fig. 2a–d, f, Supplementary Figs. 8 and 9. The confocal laser scanning microscopy (CLSM) images of different samples were repeatedly taken for at least two times, which all showed similar results as demonstrated in Fig. 6b–f, Supplementary Figs. 33–39, 41–43.

## Reporting summary

Further information on research design is available in the Nature Portfolio Reporting Summary linked to this article.

## Data availability

The source data generated in this study are provided in the 'Source Data' file. Source data are provided with this paper.

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

## Acknowledgements
The authors acknowledge financial support from the National Natural Science Foundation of China (grant no. 22005355 to W.-Q.W.), Guangdong Basic and Applied Basic Research Foundation (grant no. 2022A1515010282 to W.-Q.W.). B.C. acknowledges financial support from the Natural Science Foundation of Shanghai, China (grant no. 20ZR437400 to B.C.). We thank professor Mei Pan and professor Wei-Xiong Zhang for their help in wavelength-dependent 2PEF spectra and SHG signal testing, respectively.

## Author contributions
W.-Q.W., B.C., M.X., and T.T. designed the experiment. T.T., Y.F., and S.Z. carried out the fabrication and characterization of

DAST@HPβCD materials. T.T. and W.W. carried out density-functional theory (DFT) and natural transition orbitals (NTO) calculations. T.T., C.X., Y.C., Y.T., B.C., and W.-Q.W. carried out the PL/NLO property testing and analyzed the PL/NLO property data. T.T., Y.F., M.Y., and M.X. completed the one/two-photon in vivo bioimaging test. Y.F., T.T., and W.-Q.W. wrote the manuscript. All the authors reviewed the manuscript.

## Competing interests

The authors declare no competing interests.
