## [Peer Review File · Nature Communications]

Durable organic nonlinear optical membranes for thermotolerant lightings and in vivo bioimagingREVIEWER COMMENTS

Reviewer #1 (Remarks to the Author):

This paper explores the use of non-linear optical (NLO) materials in membranes and fibres toward applications in light-emitting diodes and biological imaging. Supramolecular control of a host-guest system herein resulted in an optoelectronic material with a high quantum yield composed of 4-N, N-dimethylamino-4'-N'-methyl-stilbazolium tosylate (DAST)@cyclodextrin (HP β CD). Importantly, DAST crystals are known to exhibit low quantum yield and can be challenging to work with and synthesize. The authors propose that by imparting geometric constraints on the NLO material, aggregation-caused quenching is reduced as is the conformational rearrangement to the twisted intramolecular charge transfer state. The single- and multi-photon photophysical properties are investigated and the electrospun materials are characterized by a variety of techniques with insight into their use as light-emitting membranes and cellular imaging materials in vitro. This work is relevant to a wide audience fitting for the readership of Nature Communications; however, several points must be addressed for this work to be published. These comments are listed below.

- The authors note on page 12 that, "Subsequent densification process further boosted the PLQY to a benchmark value of 73.5% for the DAST@HP β CD thin film (Fig. 2d, Supplementary Fig. 14 and Table 1)." This value is not shown in Table 1 and should be added. Further, it appears that only entries for the crystals and fibres are included— a row for the more dense membranes should be shown here for reference.

- Please clarify what is meant by "monomer orange fluorescence" on page 12. Does this refer to DAST by itself?

- The authors indicate that the thermotolerance is superior to other ICT materials but provide only a single reference to room-temperature phosphorescence design. Additional clarification should be added for how much more superior these materials are and reference should be made to other specific ICT materials.

- The authors claim that a working current of 100 mA in their material is "much superior" to those reported in perovskite-based materials, but the reference listed shows working currents of up to 300 mA for the reported halide perovskite solids therein. (Adv. Mater. 2020, 32, 2002495). Clarification is required, as the current claim is confusing and possibly incorrect.

- The authors claim that the long-term observation of the fluorescently labelled E. coli cultures suggests low cytotoxicity implicitly. Further experiments must be conducted and included in the main text to support this claim; at minimum this should include cell viability assays with increasing concentration of the fibres. Furthermore, it would be interesting to know how the in vitro fluorescence intensity changes as a function of time, particularly if the dextran molecules are consumed as suggested.

- (Fig. 3b) To better compare the presence/absence of emission from the TICT state in the DAST@HP β CD material (expected \sim 746 nm), the authors should include the photoluminescence profile at wavelengths above the current cut-off of 720 nm.

- On page 21 it is stated that, “the possible conveyance of the fluorescence capacity to the next-generation cells was assumed through either genetic inheritance or fluorescent fiber distribution during mitosis.” However, it would be highly unusual to have genetic inheritance of a consumed material. The authors must provide direct evidence for this claim and/or references to any other instances of this happening.

- On page 4/5, it is stated that 50% of PL intensity is maintained after 973 hours of irradiation, but it is not clear if this refers to the UV light or femtosecond pulsed laser irradiation.

- The authors claim that the DAST@HP β CD membrane exhibits a record high PLQY of 73.5 %. The previous record should be cited and included in the main text.

Reviewer #2 (Remarks to the Author):

DAST is one of the most important organic nonlinear optical (NLO) materials. In this work, DAST@HP β CD host-guest supramolecular complex was prepared by a simple method, where HP β CDs were used as the host matrices for spatially separating and suppressing the aggregation of DAST molecules. The optical properties of the as-prepared material, such as the photoluminescence quantum yield (PLQY), SHG, luminescence, and vivo bioimaging were investigated. In overall, valuable information has been presented in this manuscript. But some descriptions are unsuitable or unclear, at least in this version. So, to improve the quality, so that the readers from different fields can better understand their descriptions, it is recommended to make further revisions. Herein, my comments are:

(1) Lines 1-2 of page 3, the authors claimed that “the DAST molecules suffer from notorious aggregation caused quenching (ACQ)”. However, to the best of our knowledge, DAST might be an aggregation-induced emission material. So, please check and make suitable revision;

(2) The 2nd paragraph of page 3, it was said that “Motivated by previous works, DAST materials with SHG and two-photon-excited fluorescence (2PEF) properties, namely, utilize low-energy visible/NIR excitation through MPA pathway to produce high-energy emissions, could be used as promising fluorescence probes for realizing real-time bioimaging with high spatial resolution, as long as their intramolecular noncentrosymmetric arrangement could be enduringly fixed and their photostability could be significantly enhanced”. Please note that 2PEF/3PEF is third-order nonlinear optical behavior, which is not related to the noncentrosymmetric arrangement of DAST. So, this description is unsuitable;

(3) Lines 19-21 of page 4, the authors said that “DAST@HP β CD membrane showed a \sim 81-fold enhancement in PLQY, namely, from 0.9% to 73.5%, which represented the best luminescent performance ever for the NLO materials”. Herein, the value of 73.5% is not the highest PLQY, and it needs to point out that the single-photon excited fluorescence is a linear optical effect;

(4) Lines 21-23 of page 4, the authors claimed that “the DAST@HP β CD membranes are not easily decomposed and deactivated in high humidity conditions or even being directly immersed in water for more than 4000 hours”. However, according to Fig. 1 and Supplementary Fig. 5, DAST is directly exposed to air or water, so it seems that DAST will inevitably react with H₂O, leading to the formation of hydrate. Consequently, SHG signal will be decreased. So, one wonder how to avoid the reactions of DAST molecules with H₂O in this case?

(5) About the IR results: the authors only pay attention to the IR peaks at 1725 and 1692 cm⁻¹, which was assigned to the esterification reaction between hydroxyl groups of HP β CD and carboxyl moieties of BTCA. However, it is expected that the reactions between DAST and HP β CD might play a more important role for the effects on the optical properties. Unfortunately, this was ignored in this version. Please note and describe more about the latter reactions;

(6) About the NMR results: 1) According to Supplementary Table 2, shift of H₆ > H₂ in DAST, why and what this suggests? 2) H₁ in DAST is lacked, please mention it in the next version; 3) Why both downfield shift for DAST and HP β CD? Briefly, the descriptions about the IR and ¹H NMR results cannot support the authors’ claim that in the DAST@HP β CD membranes, the preferential insertion of pyridine group into the cavity of HP β CD host;

(7) 2nd paragraph of page 8, the authors said that “the copolymerization process triggered among numerous hydroxyl groups”. The question is what are the effects of H₂O in this case? And why and how H₂O trigger the copolymerization? Could you show the related reactions?

(8) 2nd paragraph of page 10, it was said that “It shows that the “hole” and “particle” of S₁ state are dispersed on the DAST molecule (Fig. 2g), implying that, ideally, the DAST molecule could achieve LE state-dominant characteristic.” Please explain how to achieve LE in this case? On the other hand, since the DFT calculated results could be similarly observed in all materials, whether one might incorrectly deduce that each one has LE state-dominant characteristic?

(9) The last on page 10, please show the exact values about the Stokes-shifts of DAST and DAST@HP β CD, respectively;

(10) Decrease of the Stokes-shift will lead to the increase of the self-absorption effect, different from the description on page 11?

(11) On page 11, it was claimed “C-H... π , hydrogen bond interactions”. However, in chemistry, it is hard to form the C-H... π hydrogen bond. Please note this;

(12) Line 1 of page 12, what will be formed by DAST and at the host-guest stoichiometric ratio of 12:1? In other words, please describe or illustrate the structure in this case. Similarly, it is hard to understand the structure formed at “the host-guest stoichiometric ratios of ~109:1”, as mentioned on page 15;

(13) About the DSC results: 1) the last of page 12, for the DAST@HP β CD, it was said that “can tolerate a high temperature of up to 300 oC without obvious decomposition”. However, the highest temperature in Supplementary Fig. 18b is only 250 oC? 2) Please assign the peak at 67.8 oC in Supplementary Fig. 18b;

(14) Lines 11-15 of page 13, it was said that “It is worth noting that the TICT state in DAST sample is easily excited within tens of fs due to the low energy barrier. The absorption peaks experienced significant blue-shifts when gradually increasing the delay time from 10 fs to 1 ps (Fig. 3e), suggesting that the DAST molecules could freely relax from the LE state to the TICT state owing to the small energy barrier, especially under high solution polarity environment”. Why? Whether this phenomenon might be

induced by other factors? For example, by the bandgap renormalization?

(15) For Fig 3c and d, what are these two spectra referred to? ΔT or ΔA , $\Delta T/T$?

(11) Page 15, please explain how “the crosslinked HP β CD matrix” affect the reactions of DAST with H₂O?

(12) Comparison of Fig. 5a and Supplementary Fig. 28 reveals that SHG signal significantly decreased after DAST had been soaked in H₂O. This further suggests the chemical reactions between DAST and H₂O, as mentioned above. So, the related descriptions are needed to be revised. Please note this;

(16) Lines 2-4 of page 20. The authors claimed that “Note, at the excitation wavelength of 488 nm, one-photon excited green fluorescence was captured, while at the excitation wavelength of 1000 nm, two-photon excited orange fluorescence was captured.” However, numerous previous works reported that the color of linear fluorescence of DAST is orange (e.g. see Bezkravnaya O.N., et al., Journal of Non-Crystalline Solids, 535, 119957 (2020)), why it is green in your work? As we know, the colors of linear fluorescence, 2PEF, and 3PEF generally are very close or similar (e.g. see Wang Y., et al., Nano Letters, 16, 448-453 (2016)), why they are so different in your measurements?

(17) References: 1) 28 and 29, pages are lacked; 2) 59, the journal name is incomplete;

(18) Typing errors: 1) Page 5, “It is worth pointing out the electrospinning process...” can be revised to “It is worth pointing out that the electrospinning process”; 2) Page 6, it was described that “DAST easily decomposes in water...”. It seems that the “decompose” can be revised to “dissolve”; 3) Lines 8-9 of page 13, “two samples” can be revised to “two samples”; 4) Page 15, it was said that “0.003 W (i.e. a current of 0.1 mA and a voltage of 3 V)”. Whether “0.003 W” be revised to “0.0003 W”; 5) Page 20, “Supplementary Fig. 31d-e” can be revised to “Supplementary Fig. 31d-f”.

Response to Reviewers' Comments

REVIEWER #1

This paper explores the use of non-linear optical (NLO) materials in membranes and fibres toward applications in light-emitting diodes and biological imaging. Supramolecular control of a host-guest system herein resulted in an optoelectronic material with a high quantum yield composed of 4-N, N-dimethylamino-4'-N'-methylstilbazolium tosylate (DAST)@cyclodextrin (HP β CD). Importantly, DAST crystals are known to exhibit low quantum yield and can be challenging to work with and synthesize. The authors propose that by imparting geometric constraints on the NLO material, aggregation-caused quenching is reduced as is the conformational rearrangement to the twisted intramolecular charge transfer state. The single- and multi-photon photophysical properties are investigated and the electrospun materials are characterized by a variety of techniques with insight into their use as light-emitting membranes and cellular imaging materials in vitro. This work is relevant to a wide audience fitting for the readership of Nature Communications; however, several points must be addressed for this work to be published. These comments are listed below

Response: We highly appreciated the reviewer's positive endorsement of our work. We have thoroughly revised the manuscript according to the reviewer's constructive comments.

Comment #1. *The authors note on page 12 that, "Subsequent densification process further boosted the PLQY to a benchmark value of 73.5% for the DAST@HP β CD thin film (Fig. 2d, Supplementary Fig. 14 and Table 1)." This value is not shown in Table 1 and should be added. Further, it appears that only entries for the crystals and fibers are included— a row for the more dense membranes should be shown here for reference.*

Response: We thank the reviewer's constructive advice. Accordingly, we have added a row that showed the relevant information of more dense membranes (DAST@HP β CD thin film) in the updated Table 1 of the revised manuscript.

Table 1. Optical properties and energy band parameters of DAST crystal, DAST@HP β CD fiber and DAST@HP β CD thin film.

Materials	λ_{abs} ^[a] [nm]	λ_{em} ^[b] [nm]	τ_1 /ratio [ns]	τ_2 /ratio [ns]	E_g ^[c] [eV]	HOMO ^[d] [eV]	LUMO ^[e] [eV]	PLQY [%]
DAST	578, 661	615, 746	2.284/61.49	5.529/38.51	1.61	-5.21	-3.60	0.9
DAST@HP β CD fiber	457	583	2.999/100	0/0	1.97	-5.56	-3.59	70.4
DAST@HP β CD thin film	466	575	2.878/100	0/0	2.15	-5.33	-3.18	73.5

Comment #2. Please clarify what is meant by “monomer orange fluorescence” on page 12. Does this refer to DAST by itself?

Response: We thank the reviewer for pointing this out. The "monomer orange fluorescence" of DAST@HP β CD fibers is in contrast to the “dual PL emission” for the DAST crystals, in which the former only emitted single orange fluorescence at ~ 583 nm, while the latter one exhibited dual PL peaks at ~ 615 nm and ~746 nm. All the PL emissions indeed came from the DAST material itself, but the different PL peak locations corresponded to the luminescence from different excited states. In order to avoid any misunderstanding, we have changed the “monomer” to “single” in the revised manuscript.

As can be seen on Page 13, “In sharp contrast, the DAST@HP β CD fibers only emitted single orange fluorescence at ~583 nm..., indicating the HP β CD inclusion effectively restricted the intermolecular interaction, trans–cis isomerization and forbids excimer formation of DAST molecules^{8,33}”.

Comment #3. The authors indicate that the thermotolerance is superior to other ICT materials but provide only a single reference to room-temperature phosphorescence design. Additional clarification should be added for how much more superior these materials are and reference should be made to other specific ICT materials.

Response: We thank the reviewer for pointing this out and providing the constructive suggestion. We have cited some more relevant references (Ref. 63-65) and further clarify how superior of thermal stability of DAST@HP β CD thin film is over the other

ICT materials.

As can be seen on Page 16, “Notably, even when the temperature reaches 300 °C, the PLQY of DAST@HPβCD thin film still reached 67.8% (Supplementary Fig. 23), indicating the robust thermotolerance, which is superior to other reported ICT materials⁶³⁻⁶⁵, such as organic conjugated deep-blue nano-emitters like MC8TPA and OEYTPA (thermal tolerance up to ~120 °C)⁶⁴, intermolecular coupling of OEYTPA and CmTPA materials (thermal tolerance up to ~160 °C)⁶⁵”.

Comment #4. *The authors claim that a working current of 100 mA in their material is “much superior” to those reported in perovskite-based materials, but the reference listed shows working currents of up to 300 mA for the reported halide perovskite solids therein. (Adv. Mater. 2020, 32, 2002495). Clarification is required, as the current claim is confusing and possibly incorrect.*

Response: We thank the reviewer for pointing this out. We apologized for the incorrect claim. To avoid misunderstanding, we have modified this sentence in the revised manuscript.

As can be seen on Page 17, “It is worth pointing out that our orange LEDs can operate at a high working current of up to 100 mA. ~~which is much superior to those reported phosphor based or perovskite based LEDs⁵⁸”.~~

Comment #5. *The authors claim that the long-term observation of the fluorescently labelled E. coli cultures suggests low cytotoxicity implicitly. Further experiments must be conducted and included in the main text to support this claim; at minimum this should include cell viability assays with increasing concentration of the fibres. Furthermore, it would be interesting to know how the in vitro fluorescence intensity changes as a function of time, particularly if the dextran molecules are consumed as suggested.*

Response: We thank the reviewer for raising such an interesting question and providing the constructive advice. Accordingly, we have conducted the cell viability assays with increasing concentrations of the luminescent fibers, **which showed the increased**

amount of luminescent fibers would not affect the growth and reproduction of *E. coli* (Supplementary Fig. 39, by tracing the change of optical density at a wavelength of 600 nm (OD_{600} , the characteristic absorption peak of *E. coli*)), and the resultant fluorescent labelled-*E. coli* showed comparable or even better fluorescence performance (Supplementary Fig. 40-42). These results suggested the low cytotoxicity of DAST@HP β CD fibers. The relevant discussion has been added to the revised manuscript.

Supplementary Fig. 39. The impact of different amounts of DAST@HP β CD fibers (as indicated) on the growth performance (the change of OD_{600} over time) of *E. coli*.

Supplementary Fig. 40. The CLSM images of 20mg DAST@HP β CD fibers-stained *E. coli* taken every second using bright-field mode.

Supplementary Fig. 41. The CLSM pseudo-colored images at 1PEF of 20 mg DAST@HPβCD fibers-stained *E. coli* taken every second at 488 nm excitation.

Supplementary Fig. 42. The CLSM pseudo-colored images at 2PEF of 20 mg DAST@HPβCD fibers-stained *E. coli* taken every second at 1000 nm excitation.

As can be seen on Page 23, “To investigate the possible cytotoxicity of DAST@HPβCD fibers on the *E. coli*, we conducted the cell viability assays with increasing concentrations of the luminescent fibers, and evaluated its impact on the growth of *E. coli* (by tracing the change of optical density at a wavelength of 600 nm (OD_{600} , the characteristic absorption peak of *E. coli*)) and the fluorescence performance of labeled *E. coli*. Typically, 10 ml of alive *E. coli* was incubated with 0-20 mg of DAST@HPβCD fibers. As a result, there was almost no difference in the growth of *E. coli*, regardless of different concentrations of DAST@HPβCD fibers (Supplementary Fig. 39). The *E. coli* labeled with an increased amount of DAST@HPβCD fibers (i.e. 20 mg fibers) exhibited brighter fluorescence, either excited at 488 nm or 10000 nm

(Supplementary Fig. 40-42). These results suggested the low cytotoxicity of DAST@HP β CD fibers, which are promising to be used as biological fluorescent labeling reagent.”

As for the question “*how the in vitro fluorescence intensity changes as a function of time, particularly if the dextran molecules are consumed as suggested*”, we would like to point out that though the dextran molecules are the food source of *E. coli*, **it is the DAST@HP β CD fibers entirety (including the luminescent components), rather than the HP β CD only**, being consumed by the *E. coli*. As can be shown in Fig. 6b and 6f, the fluorescence intensity is even stronger after the *E. coli* was incubated with the DAST@HP β CD fibers for 96 hours (under the circumstance of when all the DAST@HP β CD fibers were consumed by the *E. coli*), similar to the abovementioned effect of feeding the *E. coli* with high dosage of DAST@HP β CD fibers. One can also image the increased portions of fluorescent fragments were confined in a narrower space, and a brighter fluorescence can be expected. It is also worth pointing out the DAST@HP β CD fibers soaked in water (without consumption by *E. coli*) can also preserve their fluorescence brightness when continuously irradiated by 488 nm and 1000 nm laser for 30 mins (Supplementary Fig. 32), which in turn, indicates the negligible change of fluorescence intensity as a function of time. We also observed extensive amount of fluorescent *E. coli* cells in the medium without observable fluorescence quenching after a fortnight fed-batch experiment (Supplementary Fig. 38), also suggesting the negligible fluorescence intensity changes as a function of time. Unfortunately, due to the limitation of our instrument, it is difficult for us to directly quantify the actual fluorescence intensity when we took the CLSM images, though we also believed such information can give us more comprehensive details and more direct comparison. We are still looking for a cooperation partner who has a more advanced facility to quantify the change of fluorescence intensity, and hopefully, this kind of systematic study will be reported in our follow-up work.

Fig. 6 (b) The CLSM images of *E. coli* incubated with the DAST@HP β CD fibers for (b) 8 h and (f) 96 h at 1000 nm excitation (pseudo-colored images).

Supplementary Fig. 32. The confocal laser scanning microscopy (CLSM) images of DAST@HP β CD fibers soaked in water were taken immediately (a-c), over a period of 15 min (d-f), and over a period of 30 min (g-i), in which the images were taken at bright-field mode (left); 488 nm excitation (green color, pseudo-colored image) and 1000 nm excitation (orange color, pseudo-colored image).

Supplementary Fig. 38. The CLMS images of DAST@HPβCD-stained *E. coli* cells which were continuously fed with LB medium for 14 days, and the images were taken at bright-field mode (left); 488 nm excitation (green color, pseudo-colored image) and 1000 nm excitation (orange color, pseudo-colored image).

Comment #6. (Fig. 3b) To better compare the presence/absence of emission from the TICT state in the DAST@HPβCD material (expected ~746 nm), the authors should include the photoluminescence profile at wavelengths above the current cut-off of 720 nm.

Response: We thank the reviewer for the valuable suggestion. Accordingly, we have provided the photoluminescence profile at wavelengths above 720 nm to better compare the presence/absence of emission from the TICT state in the DAST@HPβCD material. As can be seen below updated Fig. 3b, it did not show the emission from the TICT state expected at ~ 746 nm.

Fig. 3b Temperature-dependent PL intensity mappings of DAST@HPβCD thin film.

Comment #7. *On page 21 it is stated that, “the possible conveyance of the fluorescence capacity to the next-generation cells was assumed through either genetic inheritance or fluorescent fiber distribution during mitosis.” However, it would be highly unusual to have genetic inheritance of a consumed material. The authors must provide direct evidence for this claim and/or references to any other instances of this happening.*

Response: We thank the reviewer for pointing this out. Indeed, as stated in the main text, genetic inheritance is one of the assumptions in this study and no relevant references can support this claim. Considering the investigation of genetic inheritance is out of the scope of this study, we have removed this description in the revised manuscript to avoid any misunderstanding.

*As can be seen on Page 23, “Considering an extensive amount of fluorescent *E. coli* cells were found in the medium, the possible conveyance of the fluorescence capacity to the next-generation cells was assumed through ~~either genetic inheritance or~~ fluorescent fiber distribution during mitosis, which could be further studied in the future.”*

Comment #8. *On page 4/5, it is stated that 50% of PL intensity is maintained after 973 hours of irradiation, but it is not clear if this refers to the UV light or femtosecond pulsed laser irradiation.*

Response: We thank the reviewer for pointing this out. In the revised manuscript, we have specified this information.

As can be seen on Page 4-5, “Notably, the DAST@HP β CD membranes are extremely stable under ultraviolet (UV) light (a power density of 120 mW/cm²) or femtosecond laser (a wavelength of 1000 nm, 140 fs/pulse, 80 MHz repetition), which can preserve 50% of initial PL intensity after continuous UV light irradiation for 974-hour.”

Comment #9. *The authors claim that the DAST@HP β CD membrane exhibits a record high PLQY of 73.5%. The previous record should be cited and included in the main text.*

Response: We thank the reviewer for pointing this out. Accordingly, the previous records about the PLQYs of DAST and its derivatives were cited as Ref. 8 and 33, and discussed in the main text of revised manuscript.

As can be seen on Page 4, “As a result of spatial confinement, geometric restriction and intra-/inter-molecular packing modulation, the DAST@HP β CD membrane showed a ~81-fold enhancement in PLQY, namely, from 0.9% to 73.5%, which represented the best 1PEF performance for the DAST-based materials and their derivatives^{8,33}”.

REVIEWER #2

DAST is one of the most important organic nonlinear optical (NLO) materials. In this work, DAST@HP β CD host-guest supramolecular complex was prepared by a simple method, where HP β CDs were used as the host matrices for spatially separating and suppressing the aggregation of DAST molecules. The optical properties of the as-prepared material, such as the photoluminescence quantum yield (PLQY), SHG, luminescence, and vivo bioimaging were investigated. In overall, valuable information has been presented in this manuscript. But some descriptions are unsuitable or unclear, at least in this version. So, to improve the quality, so that the readers from different fields can better understand their descriptions, it is recommended to make further revisions.

Response: We highly appreciated the reviewer's positive recognition of our work. Based on the reviewer's constructive comments, we have made further revisions to improve the quality and accessibility of the content for the readers from different fields.

Comment #1. *Lines 1-2 of page 3, the authors claimed that "the DAST molecules suffer from notorious aggregation caused quenching (ACQ)". However, to the best of our knowledge, DAST might be an aggregation-induced emission material. So, please check and make suitable revision;*

Response: We thank the reviewer for bringing this to our attention. We would like to point out that there are several literatures stating that the DAST is indeed **not** an aggregation-induced emission material (*Nanoscale* 8, 18882-18886 (2016); *Nanoscale Res. Lett.* 14, 269 (2019); *Chem. Rev.* 103, 3899-4032 (2003); *J. Phys. Chem. A* 115, 10689-10697 (2011) and *ACS Photonics* 7, 2132-2138 (2020)). Although DAST has a stilbene-like structure, and the stilbene dyes are well-known to be "solid-state luminescence enhancement" materials (*Adv. Optical Mater.* 9, 2002251(2021)), however, for the DAST molecules, the aggregation of them will dramatically decrease their fluorescence. As we mentioned in the main text, the DAST molecules easily suffer

from notorious aggregation-caused quenching (ACQ). This claim can be well-supported by the fact that the PLQY of DAST crystals (normally forming dimers, aggregates or excimers between neighboring crystals) is only 0.9 % (*ACS Photonics* 7, 2132-2138 (2020)), while the PLQY of DAST monomer dispersed in methanol solution (largely preserved the single-molecule state) is increased to 9.22%. Similarly, the ACQ effect was also observed in other similar organic crystals, such as ‘trans-4-[4-(dimethylamino)styryl]-1-methylpyridinium iodide (DASPI, with similar molecular structure to DAST) (*J. Am. Chem. Soc.* 138, 1118-1121 (2016)). For our fabricated DAST@HP β CD, the supramolecular system of HP β CD clusters can effectively isolate DAST molecules and suppress their aggregation via robust host-guest inclusion, thus restraining the aggregation-caused quenching and yielding high PLQY >70%. We hope this explanation would be clear enough to address the concern from the reviewer.

Comment #2. *The 2nd paragraph of page 3, it was said that “Motivated by previous works, DAST materials with SHG and two-photon-excited fluorescence (2PEF) properties, namely, utilize low-energy visible/NIR excitation through MPA pathway to produce high-energy emissions, could be used as promising fluorescence probes for realizing real-time bioimaging with high spatial resolution, as long as their intramolecular noncentrosymmetric arrangement could be enduringly fixed and their photostability could be significantly enhanced”. Please note that 2PEF/3PEF is third-order nonlinear optical behavior, which is not related to the noncentrosymmetric arrangement of DAST. So, this description is unsuitable;*

Response: We thank the reviewer for pointing this out. Just as you said only the SHG property of DAST required noncentrosymmetric arrangement, to clarify it, we will revise the description in the second paragraph of page 3 to accurately reflect the properties of DAST materials with SHG and two-photon-excited fluorescence (2PEF). We appreciate your suggestion and thank you for helping us improve the quality of our content.

As can be seen on Page 3, “Motivated by previous works, the DAST materials with SHG and two-photon-excited fluorescence (2PEF) properties, namely, utilize low

photon-energy visible/NIR excitation through MPA pathway to produce high photon-energy emissions,^{3,30} showcased great potential to be used as fluorescence probes for realizing real-time bioimaging with high spatial resolution. ~~s long as their intramolecular noncentrosymmetric arrangement could be enduringly fixed and their photostability could be significantly enhanced.~~

Comment #3. *Lines 19-21 of page 4, the authors said that “DAST@HPβCD membrane showed a ~81-fold enhancement in PLQY, namely, from 0.9% to 73.5%, which represented the best luminescent performance ever for the NLO materials”. Herein, the value of 73.5% is not the highest PLQY, and it needs to point out that the single-photon excited fluorescence is a linear optical effect;*

Response: We appreciate the reviewer’s help in improving the clarity and accuracy of our description. The NLO materials such as DAST have normally been employed as optoelectronic frequency conversion materials (utilizing the SHG or 2PA property), rather than the main luminescent materials that worked at 1PEF. Hence, the PLQYs of these NLO materials are not necessary very high and not the main focus. In our case, we found the DAST@HPβCD membrane showed a high PLQY of 73.5%, though this is related to linear optical effect, it can be used for lightings and patterned displays application (Figure 4 of manuscript). In addition, the high PLQY can guarantee a good imaging effect as it influences the two-photon action cross-section, which is an important parameter for achieving high-resolution two-photon or multi-photon imaging. To the best of our knowledge, the PLQY of 73.5% for the DAST@HPβCD membrane represented the best 1PEF performance for the DAST materials and their derivatives. To avoid any misunderstanding and further clarify this claim, we have modified this sentence in the revised manuscript.

As can be seen on Page 4, “As a result of spatial confinement, geometric restriction and intra-/inter-molecular packing modulation, the DAST@HPβCD membrane showed a ~81-fold enhancement in PLQY, namely, from 0.9% to 73.5%, which represented the best 1PEF performance for the DAST-based materials and their derivatives^{8,33}.”

Comment #4. *Lines 21-23 of page 4, the authors claimed that “the DAST@HPβCD membranes are not easily decomposed and deactivated in high humidity conditions or even being directly immersed in water for more than 4000 hours”. However, according to Fig. 1 and Supplementary Fig. 5, DAST is directly exposed to air or water, so it seems that DAST will inevitably react with H₂O, leading to the formation of hydrate. Consequently, SHG signal will be decreased. So, one wonder how to avoid the reactions of DAST molecules with H₂O in this case?*

Response: We thank the reviewer for raising such an interesting question. We would like to point out that, though we electrospun the DAST@HPβCD fibers in ambient air, the preparation of electrospinning ink (the mixtures of DAST, HPβCD and BTCA dissolved in N,N-Dimethylformamide (DMF)) was conducted in a N₂-filled glovebox (with H₂O<0.01 ppm and O₂<0.01 ppm). In this case, the reactions of DAST molecules with H₂O will be avoided in the first place. More importantly, as is mentioned in the manuscript, the HPβCD molecule has a hydrophobic inner cavity (ca. 0.78 nm in diameter) that could sequester the DAST molecules (ca. 0.58 nm in molecular width) to form DAST@HPβCD host-guest complexes. Note, excessive amounts of HPβCD materials have been employed, and the molar ratio of HPβCD host and DAST guest is approximately 109:1. In this case, it is reasonable to expect that most of the DAST molecules have been encapsulated and protected by the HPβCD clusters, thus being able to against water invasion, once the DAST@HPβCD host-guest complexes formed via liquid-phase self-assembly. Additionally, the crosslinking HPβCD network via the esterification reaction with BTCA could further strengthen the protection effect and inhibit the interaction of DAST molecules with H₂O. Raman spectra showed that, **in water environment**, only when the DAST is included by HPβCD, its characteristic peaks preserve and show up (Supplementary Fig. 24). Thus, we concluded that the crosslinked HPβCD matrix could achieve robust supramolecular encapsulation of DAST molecules, which effectively inhibited the decomposition of DAST materials or formation of DAST-based hydrates, and maintained the structural integrity of the DAST@HPβCD host-guest inclusions even under high humidity conditions. As for the decreased SHG signal after the DAST@HPβCD fiber was immersed in water for 4000

hours, the reasons have been explained in our detailed response to Comment #17. Upon directly immersing of DAST@HP β CD in water for such a long time (regarded as an extremely harsh condition), there will still be some chance for the breakage of crosslinking network, possibly owing to the undesirable ester hydrolysis. And in this case, the inner DAST was gradually accessible by the H₂O molecules, which will cause the decomposition of DAST and/or formation of DAST hydrate, thus gradually losing the SHG signals upon continuously extended water immersion time. Similar reduction of PL intensity by 40% has been observed for the DAST@HP β CD fibers immersed in water for 4000 hours (Supplementary Fig. 29). We hope this explanation could dismiss any confusion or concerns from the reviewer. To clarify it, we have further explained the details of how to avoid the reaction of DAST molecules with H₂O in the revised manuscript.

Supplementary Fig. 24. Raman spectra of water, DAST in water, HP β CD in water and DAST@HP β CD in water.

As can be seen on Page 5, “The electrospinning ink was prepared in **N₂-filled glovebox (with H₂O<0.01 ppm and O₂<0.01 ppm)** by simply mixing HP β CD host molecules, DAST guest molecules and BTCA crosslinking agents together according to weight ratios as described in the Methods section (Supplementary Fig. 1).

As can be seen on Page 16-17, “Specifically, the crosslinked HP β CD matrix effectively inhibited the reactions of DAST with H₂O in several ways. First of all, the HP β CD host can effectively confine the DAST molecules within the hydrophobic inner

cavities. Secondly, the HP β CD inclusion can modify the local environment around DAST molecules and impart geometric constraints, which reduced the reactivity of the DAST with H₂O. Last but not least, the crosslinked HP β CD matrix that exposed hydrophobic moieties as much as possible could serve as a robust protection barrier to prevent potential water invasion to the encapsulated DAST molecules, thus reducing the likelihood of decomposition or other unwanted reactions.”

As can be seen on Page 18, “Raman spectra showed that, in water environment, only when the DAST is included by HP β CD, its characteristic peaks preserve and show up (Supplementary Fig. 24). This result confirmed the crosslinked HP β CD matrix could achieve robust supramolecular encapsulation of DAST molecules, which prevented them from decomposition or forming hydrates even in high humidity conditions.”

Comment #5. *About the IR results: the authors only pay attention to the IR peaks at 1725 and 1692 cm⁻¹, which was assigned to the esterification reaction between hydroxyl groups of HP β CD and carboxyl moieties of BTCA. However, it is expected that the reactions between DAST and HP β CD might play a more important role for the effects on the optical properties. Unfortunately, this was ignored in this version. Please note and describe more about the latter reactions.*

Response: We thank the reviewer for pointing this out and providing constructive advice. We agree that the reactions between DAST and HP β CD play a more important role for the effects on the optical properties. Thus, the reactions between functional groups of the HP β CD and DAST have been studied by Fourier transform infrared (FTIR) spectroscopy. The broad vibrational bands in the range of 3000–3500 cm⁻¹ can be assigned to O–H stretching vibrations (3391 cm⁻¹) of HP β CD and N–H stretching vibrations (3431 cm⁻¹) of DAST. The vibration bands at 1414 cm⁻¹, 1636 cm⁻¹ and 1652 cm⁻¹ can be ascribed to C–N stretching vibrations and N–H bending vibrations of DAST, as well as O–H bending vibrations of HP β CD, respectively (Supplementary Fig. 3c and d). After forming DAST@HP β CD complex, one could observe the intensities of both the O–H/N–H stretching vibrations around ~3400 cm⁻¹ and the O–H/N–H bending vibrations around ~1640 cm⁻¹ increased (Supplementary Fig. 3c), and the sharp

decrease of C–N stretching vibrations (Supplementary Fig. 3d). These peak intensity changes suggested that there may be strong hydrogen bonding interactions between DAST and HP β CD, which is advantageous for suppressing the trans–cis isomerization of DAST and thus enhancing its PLQY. Accordingly, the relevant discussion also be added in the revised version.

As can be seen on Page 6, “We also investigated the reactions between the functional groups of HP β CD and DAST via FTIR study. The broad vibrational bands in the range of 3000–3500 cm⁻¹ can be assigned to O–H stretching vibrations (3391 cm⁻¹) of HP β CD and N–H stretching vibrations (3431 cm⁻¹) of DAST. The vibration bands at 1414 cm⁻¹, 1636 cm⁻¹ and 1652 cm⁻¹ can be ascribed to C–N stretching vibrations and N-H bending vibrations of DAST, as well as O–H bending vibrations of HP β CD, respectively (Supplementary Fig. 3c and d). After forming DAST@HP β CD complex, one could observe the intensities of both the O–H/N–H stretching vibrations around ~3400 cm⁻¹ and the O–H/N–H bending vibrations around ~1640 cm⁻¹ increased (Supplementary Fig. 3c), and the sharp decrease of C–N stretching vibrations (Supplementary Fig. 3d). These peak intensity changes suggested that there may be strong hydrogen bonding interactions between DAST and HP β CD, which is advantageous for suppressing the trans–cis isomerization of DAST and thus enhancing its PLQY.”

Supplementary Fig. 3. (a) FTIR spectra of HP β CD, BTCA and HP β CD@BTCA samples. (b) The zoom-in view of (a) within a wavenumber range from 1400 cm^{-1} to 2100 cm^{-1} . (c) FTIR spectra of DAST, HP β CD and DAST@HP β CD samples. (d) The zoom-in view of (c) within a wavenumber range from 1200 cm^{-1} to 1600 cm^{-1} .

Comment #6. *About the NMR results: 1) According to Supplementary Table 2, shift of $H_6 > H_2$ in DAST, why and what this suggests? 2) H_1 in DAST is lacked, please mention it in the next version; 3) Why both downfield shift for DAST and HP β CD? Briefly, the descriptions about the IR and ^1H NMR results cannot support the authors' claim that in the DAST@HP β CD membranes, the preferential insertion of pyridine group into the cavity of HP β CD host.*

Response: We thank the reviewer for pointing this out. We have re-analyzed the NMR data and relabeled the corresponding H protons, for instance, the H_2 was relabeled as H_1 for the DAST sample, and now all the H protons were correctly assigned and identified, as can be seen in below updated Fig. 1b.

Specifically, the shift of $H_6 > H_1$ (previous H_2) in DAST can be explained by the fact that the H_6 proton residing on the C=C bond of DAST, which serves as the bridge for connecting the conjugated large π bond, so it is more active and easier be affected than

the H₁ proton residing around the dimethylamino group. Hence, one could expect more robust interactions between H₆ in DAST with HPβCD.

The shifts (either downfield or highfield shift) of the ¹H NMR signals of both DAST and HPβCD for the DAST@HPβCD composites are associated with the different disturbances of electronic environment upon the insertion of DAST into the HPβCD cavity. Due to different affinity between HPβCD cavity and different functional groups (i.e. the hydrophobic dimethylamino group and hydrophilic pyridine ring) of the DAST molecule, the strengthened anisotropic effect has been observed, in which the different protons residing on different spatial locations of aromatic rings and C=C bonds in DAST would encounter with either shielding or deshielding effect (*Chem. Rev*, 98, 1755-1786 (1998)), thus resulting in the chemical shifts of protons to different directions. Accordingly, we have modified the relevant discussion, especially further clarifying the insertion mode of DAST molecule into the cavity of HPβCD in the revised manuscript.

As can be seen on Page 7, “After forming the DAST@HPβCD complex, the resonances of H₁, H₂, H₅, H₆ and H₇ protons in DAST molecule showed highfield shift, while the resonances of H₃, H₄ and H₈ protons in DAST molecule experienced downfield shifts (Fig. 1b and Supplementary Table 2). This can be explained by the anisotropic effect of aromatic rings and the C=C bonds in DAST molecule during NMR characterization, in which the different protons residing on different spatial locations would encounter either shielding or deshielding effect⁵⁰, thus resulting in the chemical shifts of protons to different directions. The different affinity between HPβCD cavity and different functional groups (i.e. the hydrophobic dimethylamino group and hydrophilic pyridine ring) of the DAST molecule may also contribute to the abovementioned anisotropy effect. As for the resonances of H₂, H₃, H₄, H₅ and H₆ protons residing on the cavity of the HPβCD molecule, they all highfield shifted after DAST insertion (Supplementary Table 3). Overall, the shifts in the ¹H NMR spectra of both DAST and HPβCD after forming DAST@HPβCD composites were related to the hydrogen bonding interactions and electrostatic interactions between these molecules.⁵¹ In addition, the correlation spectroscopy (COSY) ¹H NMR was measured to gain more

insight into the geometric structure of the DAST@HP β CD complex. Specifically, strongly coupled signals between both H₁ and H₂ protons of DAST with the protons in HP β CD further confirmed the insertion of DAST molecule into the cavity of HP β CD host. Overall, combining the ICD signal, ¹H NMR and COSY NMR results, we confirmed that the DAST molecules readily reside in the HP β CD cavities to form the DAST@HP β CD supramolecular host-guest inclusion complexes.”

Fig. 1. (b) The ¹H NMR spectra (600 MHz, D₆MSO) of DAST, HP β CD and DAST@HP β CD. (c) The COSY NMR of DAST@HP β CD.

Comment #7. 2nd paragraph of page 8, the authors said that “the copolymerization process triggered among numerous hydroxyl groups”. The question is what are the effects of H₂O in this case? And why and how H₂O trigger the copolymerization? Could you show the related reactions?

Response: We thank the reviewer for raising these questions. According to the previous reports ((*J. Pharm. Sci.* 105, 2556-2569(2016); *Int. J. Pharm.* 560, 228-234 (2019))), in the high humidity condition, the -OH groups of cyclodextrin (CD) are capable of forming hydrogen bonds with surrounding H₂O molecules, which would facilitate the self-assembly of CDs into aggregates or clusters.

In our study, there are numerous HP β CD molecules assembled on the surface of nanofiber skeleton. When the nanofibers are exposed in high humidity condition, hydrogen bondings between HP β CD and H₂O molecules would be formed. In addition to interacting with HP β CD, the H₂O also acts as a "flowable medium", which could gradually infiltrate to the voids of the nanofibers and provide a favorable driving force

for facilitating the self-assembly and aggregation of HP β CD molecules. In this case, the numerous H₂O-interacted HP β CD molecules on the surface of nanofibers tend to merge together, which promoted copolymerization between adjacent fibers. As a result, the voids and gaps between the nanofibers were filled, and a transparent DAST@HP β CD thin film was formed. Accordingly, we have explained the effects of H₂O, as well as why and how H₂O triggers the copolymerization process in the revised manuscript.

As can be seen on Page 9, “Interestingly, upon exposure of DAST@HP β CD fibers in a high relative humidity (RH) environment (~70-85%) for 10 mins, it turned into a transparent thin film with extremely dense morphology, which is possibly due to the hydrogen bonding interactions and copolymerization process triggered in the presence of H₂O (Fig. 2c and 2d). In this case, the H₂O could not only interact with HP β CD oligomers assembled on the surface of fibers via hydrogen bonding, but also facilitate the self-assembly and/or aggregation of numerous HP β CD molecules (Supplementary Fig. 7)⁵², which ultimately promoted copolymerization/interaction among adjacent fibers (i.e. hydrogen bonding between -OH groups of HP β CD molecules). As a result, all of the pores and voids within the interconnected network were filled out and the pristine porous membrane was densified to form a compact and transparent thin film (Supplementary Fig. 8).”

Supplementary Fig. 7. Schematic illustration of the hydrogen bonding interactions and self-assembly process of forming densified DAST@HP β CD materials under high humidity condition.

Comment #8. 2nd paragraph of page 10, it was said that “It shows that the “hole” and “particle” of S1 state are dispersed on the DAST molecule (Fig. 2g), implying that, ideally, the DAST molecule could achieve LE state-dominant characteristic.” Please explain how to achieve LE in this case? On the other hand, since the DFT calculated results could be similarly observed in all materials, whether one might incorrectly deduce that each one has LE state-dominant characteristic?

Response: We thank the reviewer for pointing this out. Though according to the result of NTO calculation, the DAST molecule could also achieve LE state-dominant characteristics, one should also consider the **physical form (solid or liquid), the effect of solvents with different polarity and other more complicated factors.** For instance, the LE state-dominant characteristic of DAST materials could possibly achieved by dissolving them in some kinds of solvents with specific polarity, as long as the molecular isomerization and transition to TICT state could be concurrently inhibited in the liquid state (Supplementary Fig. 10). However, this is still of great challenge to be achieved. In contrast, only one PL peak was observed around 600 nm for the DAST@HP β CD sample, regardless of in solid-state or liquid-state, which implied the HP β CD inclusion could alter the DAST molecule environment and at a large extent minimize the solvatochromic effects (Fig. R1).

Fig. R1. The PL spectra of the DAST@HP β CD fibers either in solid-state or liquid-state.

Though the DFT calculation results could be similarly observed in both DAST (LE state = 63.89 %) and DAST@HP β CD (LE state = 65.04 %), it should be noted that the calculation is based on the model of using a **single molecule (as is mentioned in the caption of Fig. 2g)**. Even based on the analysis of single molecule mode, the LE proportion in DAST@HP β CD was still slightly higher than that in DAST, so that one could still reasonably deduce the HP β CD inclusion could enhance the LE state-dominant characteristic of DAST. More importantly, in a real case, whether each one has LE state-dominant characteristics should also consider **its physical form (solid or liquid), the solvent effect, the specific molecular compositions and other more complicated factors**. Compared to the DAST@HP β CD single molecule (NTO simulation), the molar ratio of HP β CD:DAST in our fabricated DAST@HP β CD composited fibers is nearly 109:1, which could achieve enhanced nanoconfinement effect on DAST molecule, weakened solvatochromic effect and further strengthen its LE-dominant characteristics. To clarify this issue, we have modified our description in the revised manuscript.

Fig. 2g. NTO simulation of DAST and DAST@HP β CD. **Note, the calculation is based on the model of using single molecule.**

As can be seen on Page 11, “It shows that the “hole” and “particle” of S_1 state are dispersed on the DAST molecule (Fig. 2g), implying that, ideally, the DAST molecule could achieve LE state-dominant characteristic. For instance, this could possibly happen by dissolving the DAST materials in some kinds of solvents with specific polarity, as long as the molecular isomerization and transition to TICT state could be

inhibited in the liquid state (Supplementary Fig. 10). In reality, the DAST material is not efficiently luminescent and loses its NLO properties owing to its vulnerability in ambient air and/or polar solvent environment. Note, the above NTO simulation is based on the model of using a single molecule, while in a real case, whether each one has LE state-dominant characteristics should also consider its physical form (solid or liquid), the solvent effect, the specific molecular compositions and other more complicated factors.”

Supplementary Fig. 10. The PL spectra of the DAST crystals dissolved in different solvents with different polarity.

Comment #9. *The last on page 10, please show the exact values about the Stokes-shifts of DAST and DAST@HP β CD, respectively*

Response: We thank the reviewer for providing constructive advice. We regret to point out that it is difficult for us to show the exact values of the Stokes-shift, especially for the DAST sample, in which there are two PL peaks and a very broad absorption spectrum with several peak signals (Supplementary Fig. 12). Alternatively, to compare the self-absorption effect between DAST and DAST@HP β CD, one could quantify the spectral overlap area between absorption and luminescence. And the relevant discussion has been added to the revised manuscript.

As can be seen on Page 12, “ Specifically, the DAST crystal exhibited a large spectral overlap between absorption and luminescence (an overlap area of $\sim 33.22\%$). Upon the introduction of large amounts of HP β CD molecules, the absorption spectra underwent

a blue-shift along with the enhancement of absorption intensity, and a reduced spectral overlap (an overlap area of $\sim 25.10\%$) was observed, which is beneficial to mitigate the self-absorption effect.

Supplementary Fig. 12. The absorption and PL spectra of DAST crystal (blue color) and DAST@HP β CD fibers (purple color).

Comment #10. *Decrease of the Stokes-shift will lead to an increase of the self-absorption effect, different from the description on page 11?*

Response: We thank the reviewer for pointing this out. We agree that a decrease of the Stokes-shift will lead to an increase of the self-absorption effect and we apologized for the inappropriate statement in the previous version. See our response to Comment #9, due to the difficulty of quantifying the exact Stokes-shift, we did not discuss it anymore in the revised version. Alternatively, the spectral overlap area between absorption and luminescence has been quantified and compared, which showed the DAST@HP β CD has minimized the self-absorption effect.

Comment #11. *On page 11, it was claimed “C-H... π , hydrogen bond interactions”. However, in chemistry, it is hard to form the C-H... π hydrogen bond. Please note this;*

Response: We thank the reviewer for the careful review. The C-H $\cdots\pi$ is a kind of electrostatic interaction (*Mater. Chem. Front.* 5, 1418 (2021); *Adv. Sci.* 7, 2000803 (2020)), which is different from the hydrogen bond interactions. To avoid any misunderstanding, we have slightly modified this description in the revised version.

*As can be seen on Page 12, “More importantly, the supramolecular host-guest inclusion complex with abundant intermolecular interactions (i.e. **electrostatic interactions**, hydrogen bond interactions, etc)⁴⁷.....”*

Comment #12. *Line 1 of page 12, what will be formed by DAST and at the host-guest stoichiometric ratio of 12:1? In other words, please describe or illustrate the structure in this case. Similarly, it is hard to understand the structure formed at “the host-guest stoichiometric ratios of ~109:1”, as mentioned on page 15;*

Response: We thank the reviewer for pointing this out and bringing this to our attention. We apologized for the unclear writing. For our demonstrated DAST@HP β CD samples, the HP β CD serves as the host, and the DAST acts as the guest. According to the host-guest chemistry and some previous reports (*ACS Photonics* 7, 2132-2138 (2020); *J. Am. Chem. Soc.* 133, 7276-7279 (2011); *Int. J. Pharm.* 560, 228-234 (2019)) as well as considering the molecular sizes of both DAST and HP β CD, we speculated the host-guest stoichiometric ratios of HP β CD and DAST were still ranged from 1:1 to 4:1, which is able to form DAST@HP β CD host-guest inclusion complex. In our study, excess amounts of HP β CD were introduced, for instance, **12:1 or 109:1 for the molar ratios of host and guest materials. Note, not all the excessive HP β CD molecules were consumed to form DAST@HP β CD host-guest complex.** On one hand, the excessive HP β CD is aimed to ensure that the DAST molecules are encapsulated by HP β CD as much as possible. On the other hand, the inclusion reaction is reversible, and the higher concentrations of reactants (i.e. HP β CD) are advantageous for driving the reaction toward forming the DAST@HP β CD host-guest inclusion product. **More importantly, it is worth pointing out a large portion of the HP β CD molecules were consumed to form the nanofiber skeleton/backbone.** Since the description of “host-

guest stoichiometric ratio of ~109:1” is not appropriate, so we rewrote this sentence in the revised manuscript.

As can be seen on Page 13, “We attributed the remarkable fluorescence enhancement to the HP β CD introduction and inclusion, as is evidenced by the gradual amplification of PL intensity when the **molar ratio** of host molecules and guest molecules was increased from 0:1 to 12:1 (Supplementary Fig. 15).

As can be seen on Page 16, “our demonstrated DAST@HP β CD fibers or thin films are composed of an excessive amount of HP β CD host molecules and a minority of DAST guest molecules (i.e. the molar ratio of HP β CD molecules and DAST molecules was approximately 109:1, which is expected to be higher than the real host-guest stoichiometric ratio of forming DAST@HP β CD, since **not all the excessive HP β CD molecules were consumed to form DAST@HP β CD and a large portion of the HP β CD molecules were consumed to form the nanofiber skeleton/backbone).**”

Comment #13. About the DSC results: 1) the last of page 12, for the DAST@HP β CD, it was said that “can tolerate a high temperature of up to 300 °C without obvious decomposition”. However, the highest temperature in Supplementary Fig. 18b is only 250 °C? 2) Please assign the peak at 67.8 °C in Supplementary Fig. 18b;

Response: We thank the reviewer for pointing this out and providing useful suggestions. The temperature of TG measurement was extended to above 300 °C. However, for the DSC measurement, considering the DAST crystal sample already lost ~18.5% of its weight at 300 °C, so higher testing temperature is not allowed to avoid polluting the sample cavity. Accordingly, we have modified the analysis of both TG and DSC results in the revised manuscript.

As can be seen on Page 13-14, “As expected, the thermal stability of DAST@HP β CD fibers are much better than that of DAST crystal. Specifically, the DAST@HP β CD fibers can tolerate a high temperature of up to 300 °C without obvious decomposition (~ 2.9 % weight loss), while the DAST crystals witnessed ~ 18.5% weight loss at 300 °C (Supplementary Fig. 19). For the DSC results, there is an

endothermic reaction at ~ 261 °C for both samples, which referred to the melting point of DAST⁵⁷. For the peak at 133.1 °C showed in DAST@HP β CD fibers, it can be possibly assigned to the breakage of hydrogen bonds or dehydration under the temperature range of ~ 120 -150 °C (Supplementary Fig. 20).”

Supplementary Fig. 19. TG spectra of (a) DAST crystals and (b) DAST@HP β CD fibers.

Supplementary Fig. 20. The DSC spectra of (a) DAST crystals and (b) DAST@HP β CD fibers. Before measurement, the DAST crystals were stored in N₂-filled glovebox and the DAST@HP β CD fibers were stored in ambient air with a relative humidity of 65%.

Comment #14. Lines 11-15 of page 13, it was said that “It is worth noting that the TICT state in DAST sample is easily excited within tens of fs due to the low energy barrier. The absorption peaks experienced significant blue-shifts when gradually increasing the delay time from 10 fs to 1 ps (Fig. 3e), suggesting that the DAST molecules could freely relax from the LE state to the TICT state owing to the small

energy barrier, especially under high solution polarity environment”. Why? Whether this phenomenon might be induced by other factors? For example, by the bandgap renormalization?

Response: We thank the reviewer for raising such an interesting question and reminding us that bandgap renormalization would be also an important factor to induce blue-shifts of absorption peaks. For TA measurement, the DAST crystals and DAST@HP β CD thin film were both dissolved in distilled water (high solution polarity environment). It is reported that the polarity of the environment can modulate the energy barrier between LE and CT states (Fig. R2, *Adv. Optical Mater.* 2, 892-901(2014)). Due to the strong interaction between the dipole moments of the charge transfer state and polar solvents, the energy of the CT state is decreased with increasing solvent polarity, whereas the energy of the LE state remains nearly unchanged (*Chem. Rev.* 103, 3899-4032 (2003); *Phys. Chem. Chem. Phys.* 20, 7514-7522 (2018)). Hence, in our case, under high solution polarity environment, the DAST molecules could freely relax from the LE state to the TICT state owing to the small energy barrier. Increasing the delay time from 10 fs to 1 ps would excite more LE state and facilitate its transition to TICT state, which resulted in energy redistribution within different excited states and led to bandgap renormalization of DAST molecules, thus significantly blue-shifting the absorption peaks (*J. Phys. Chem. A.* 115, 8183-8196 (2011)). Accordingly, we have modified the relevant discussion in the revised manuscript.

As can be seen on Page 14, “For TA measurement, the DAST crystals and DAST@HP β CD thin films were both dissolved in distilled water (high solution polarity environment). Upon 350 nm fs-laser irradiation, the contour plots of delay time-dependent TA spectra.....It is worth noting that the TICT state in DAST sample is easily excited within tens of femtoseconds. Especially under high solution polarity environment, the DAST molecules could freely relax from the LE state to the TICT state owing to the small energy barrier⁶⁰. Increasing the delay time from 10 fs to 1 ps would excite more LE state and facilitate its transition to TICT state, which resulted in energy redistribution within different excited states and led to bandgap renormalization of DAST molecules, thus significantly blue-shifting the absorption peaks⁶¹ (Fig. 3e).”

Fig. R2. Scheme of LE and CT state mixing to form an HLCT state in different environments showing the decline of the CT state with increasing polarities (*Adv. Optical Mater.* 2, 892-901(2014)).

Comment #15. For Fig 3c and d, what are these two spectra referred to? ΔT or ΔA , $\Delta T/T$?

Response: We thank the reviewer for pointing this out. Fig. 3c and 3d are the contour plots of delay time-dependent TA spectra, which referred to the change of absorption intensity (ΔA) as a function of delay time from 0 to 75 ps, while those two spectra in Fig. 3e and 3f referred to the ΔA under four specific delay times, namely, 10 fs, 100 fs, 500 fs and 1 ps. To clarify it, we have supplemented this information in the revised manuscript.

As can be seen on Page 14, “Upon 350 nm fs-laser irradiation, the contour plots of delay time-dependent TA spectra (referred to the change of absorption intensity (ΔA) as a function of delay time from 0 to 75 ps) of both DAST and DAST@HP β CD samples exhibited broad ground-state bleaching (GSB) peaks at ~ 600 nm (Fig. 3c and 3d).”

Comment #16. Page 15, please explain how “the crosslinked HP β CD matrix” affect the reactions of DAST with H₂O?

Response: We thank the reviewer’s conducive suggestion. We explained how “the crosslinked HP β CD matrix” affects the reactions of DAST with H₂O in detail in our

revised version.

As can be seen on Page 16-17, “Specifically, the crosslinked HP β CD matrix effectively inhibited the reactions of DAST with H₂O in several ways. First of all, the HP β CD host can effectively confine the DAST molecules within the hydrophobic inner cavities. Secondly, the HP β CD inclusion can modify the local environment around DAST molecules and impart geometric constraints, which reduced the reactivity of the DAST with H₂O. Last but not least, the crosslinked HP β CD matrix that exposed hydrophobic moieties as much as possible could serve as a robust protection barrier to prevent potential water invasion to the encapsulated DAST molecules, thus reducing the likelihood of decomposition or other unwanted reactions.”

Comment #17. Comparison of Fig. 5a and Supplementary Fig. 28 reveals that SHG signal significantly decreased after DAST had been soaked in H₂O. This further suggests the chemical reactions between DAST and H₂O, as mentioned above. So, the related descriptions are needed to be revised. Please note this;

Response: We thank the reviewer for pointing this out and providing valuable suggestions. See our response to Comment #4, the chemical interactions between DAST and H₂O have been largely avoided by controlling the fabrication procedures and via robust encapsulation by crosslinked HP β CD matrix. And in this case, the SHG signal should be well-maintained even the DAST@HP β CD fibers were exposed in ambient air with considerable humidity, as long as the protection from crosslinked HP β CD matrix is well-reserved. Nevertheless, upon directly immersing of DAST@HP β CD in water for quite a long time (regarded as an extremely harsh condition), there will still be some chance for the **breakage of the crosslinking network**, possibly owing to the undesirable ester hydrolysis. And in this case, the inner DAST was gradually accessible by the H₂O molecules, which will cause the decomposition of DAST and/or formation of DAST hydrate, thus gradually losing the SHG signals upon continuously extended water immersion time. Similar reduction of PL intensity by 40% has also been observed for the DAST@HP β CD fibers immersed in water for 4000 hours owing to the abovementioned decomposition or hydration

(Supplementary Fig. 29). Note, in practical utilization of the SHG property for up-conversion application, the solid-state DAST@HP β CD sample does not need to be directly immersed in water, which is believed to be stable enough to maintain the SHG signals and exhibit a largely extended lifespan. To clarify this issue, we have modified the related description in the revised manuscript.

As can be seen on Page 21, “Though promising, one could still observe decreased SHG signal of DAST@HP β CD upon water immersion for quite a long time (i.e. 4000 hours, Supplementary Fig. 30). This can be attributed to the breakage of the crosslinking network, possibly owing to the undesirable ester hydrolysis. And in this case, the inner DAST materials were readily accessible by the H₂O molecules, which will cause the decomposition of DAST and/or formation of DAST hydrate, thus gradually losing the SHG signals upon continuously extended water immersion time. Note, in practical utilization of the SHG property for up-conversion application, the solid-state DAST@HP β CD sample does not need to be directly immersed in water, which is believed to be stable enough to maintain the SHG signals and exhibit a largely extended lifespan.”

Comment #18. Lines 2-4 of page 20. The authors claimed that “Note, at the excitation wavelength of 488 nm, one-photon excited green fluorescence was captured, while at the excitation wavelength of 1000 nm, two-photon excited orange fluorescence was captured.” However, numerous previous works reported that the color of linear fluorescence of DAST is orange (e.g. see Bezkrovnyaya O.N., et al., Journal of Non-Crystalline Solids, 535, 119957 (2020)), why it is green in your work? As we know, the colors of linear fluorescence, 2PEF, and 3PEF generally are very close or similar (e.g. see Wang Y., et al., Nano Letters, 16, 448-453 (2016)), why they are so different in your measurements?

Response: We thank the reviewer for raising this insightful question. We agree that the colors of linear fluorescence, 2PEF and 3PEF generally are very close or similar. We apologize for any confusion caused by the unclear description regarding the fluorescence color of DAST@HP β CD displayed at different excitation wavelengths.

The color differences between 1PEF and 2PEF images are because we applied the **pseudocolor staining technique**. In one/two-photon confocal laser scanning microscopy (CLSM NLO 710), pseudocolor staining is commonly used to distinguish between single-photon and two-photon imaging signals, which can highlight specific structures or features of the samples that may be difficult to distinguish in the original grayscale image (*Cell Reports* 19, 203-217 (2017); *Nat. Immunol.* 24, 664-675 (2023)). In one/two-photon CLSM, the 488 nm excitation (one-photon) and 1000 nm excitation (two-photon) are conducted in two different channels, which applied different detectors. When the detectors receive the PL intensity signals of DAST-stained *E.coli* excited by 488 nm or 1000 nm excitation, the grayscale images will be firstly produced. The grayscale image is then assigned with a color map, which maps different PL intensities and converts them to different colors. To avoid any misunderstanding, we have briefly described the pseudocolor staining of fluorescence images in the revised manuscript.

As can be seen on Page 22, “The confocal laser scanning microscopy (CLSM) images (collected at bright-field (black and white color), 1PEF at 488 nm (green color), and 2PEF at 1000 nm (orange color) of the *E. coli* after incubation with DAST@HP β CD fibers for different durations are shown in Supplementary Fig. 33. **Note, to better distinguish the 1PEF and 2PEF fluorescence images, pseudocolor staining is employed, though generally the real colors of 1PEF and 2PEF are very close or similar⁸.**”.

As can be seen on Page 24-25, “Fig. 6 Demonstration of in vivo bioimaging of *E. Coli* via DAST@HP β CD fibers labeling and corresponding CLSM images. (a) The schematic illustration of in vivo bioimaging of *E. coli* by employing DAST@HP β CD fibers. The CLSM images of *E. coli* incubated with the DAST@HP β CD fibers for (b) 8 h and (c) 24 h under 1000 nm excitation, and for 96 h under (d) bright-field mode, (e) 488 nm excitation and (f) 1000 nm excitation. **Note, the fluorescence images captured at 488 nm or 1000 nm excitation are pseudo-colored for better distinction.**”

Comment #19. References: 1) 28 and 29, pages are lacked; 2) 59, the journal name is incomplete;

Response: We thank the reviewer's careful review and for pointing these out. We have supplemented the detailed information for those references in the revised manuscript.

As can be seen on page 28, “28 Tian, R. et al. Albumin-chaperoned cyanine dye yields superbright NIR-II fluorophore with enhanced pharmacokinetics. Sci. Adv. 5, eaaw0672 (2019); 29 Xu, Z. et al. Deep-brain three-photon imaging enabled by aggregation-induced emission luminogens with near-infrared-III excitation. ACS Nano, 16, 6712-6724 (2022) and 66 Zeng, Y. et al. An exceptional thermally induced four-state nonlinear optical switch arising from stepwise molecular dynamic changes in a new hybrid salt. Angew. Chem. Int. Ed. 134, e202110082 (2022).”

Comment #20. Typing errors: 1) Page 5, “It is worth pointing out the electrospinning process...” can be revised to “It is worth pointing out that the electrospinning process”; 2) Page 6, it was described that “DAST easily decomposes in water...”. It seems that the “decompose” can be revised to “dissolve”; 3) Lines 8-9 of page 13, “two samples” can be revised to “two samples”; 4) Page 15, it was said that “0.003 W (i.e. a current of 0.1 mA and a voltage of 3 V)”. Whether “0.003 W” be revised to “0.0003 W”; 5) Page 20, “Supplementary Fig. 31d-e” can be revised to “Supplementary Fig. 31d-f”.

Response: We thank the reviewer for picking up these typing errors. We greatly appreciate your time to help in improving the quality of our manuscript. Accordingly, we have made the necessary corrections in the revised manuscript.

*As can be seen on Page 5, “ It is worth pointing out **that** the electrospinning process ...”.*

*As can be seen on Page 6, “...DAST easily **dissolves** in water...”.*

*As can be seen on Page 14, “For both **two** samples...”.*

As can be seen on Page 17, “ The orange LED can be lightened up at a low power of 0.003 W (i.e. a current of 1 mA and a voltage of 3 V was applied...”.

*As can be seen on Page 22, “ In Stage II (8 ~ 24 h), the fibers were gradually bitten and consumed by *E. coli*, as is evidenced by gradual disappearance of fluorescent fiber fragments (Supplementary Fig. 33 **d-f** and Fig. 6c).”*

REVIEWER COMMENTS

Reviewer #1 (Remarks to the Author):

The authors have thoroughly revised their manuscript in response to reviewer comments, and is now suitable for publication. As a single point of feedback, I would encourage the authors to be more concise in future response letters; their response letter was 34 pages long! I appreciate the desire to be thorough but being more concise helps reviewers greatly.

Reviewer #2 (Remarks to the Author):

In this work, DAST@cyclodextrin (HP β CD) host-guest supramolecular complex with strong fluorescence and high stability as well as some applications of the materials were reported. The authors have responded to all the comments of the reviewers one-by-one, and a major revision has been made. Most of their explanations are acceptable, at least in their cases. So, it seems that the possible publication could be considered. Before doing so, a minor revision is required:

(1) About the copolymerization triggered by H₂O: Page 9 and Fig. 2c and d, the authors claimed that this phenomenon is attributed to “the hydrogen bonding interactions and copolymerization process triggered in the presence of H₂O”. In contrast, “post-annealing treatment will block the copolymerization”. To convince the readers, the authors can provide the proof(s) for their claims.

(2) About the TEM image: 1) Whether can DAST and HP β CD be determined by TEM? If yes, please show these two components in Fig. 2f, respectively; 2) It was described on page 9 “single-crystalline DAST crystals (i.e. ~12 nm in diameter)” in Fig. 2f. In this case, how DAST was encapsulated in the CD cavity?

(3) As summarized in Table 1, all the optical properties for the DAST@HP β CD fiber and DAST@HP β CD thin film are different, e.g. absorption and PL peaks, Eg, PLQY, etc. This is interesting since both are DAST@HP β CD. Why so?

(4) About the thermotolerance: Supplementary Fig. 22f shows a black region on the left, whether some of the DAST@HP β CD thin film had been destroyed at 300 oC?

(5) About the Raman spectra in Supplementary Fig. 24: Why the peaks for DAST disappeared in H₂O, especially, for the peak at ~1190 nm?

(6) For the one-photon-excited emission, two-photon excited fluorescence (2PEF), and three-photon absorption (3PA), whether the differences between DAST and DAST@HP β CD can be summarized?

(7) It was described in both abstract and conclusion that the DAST@HP β CD thin film is stable even at 350 oC. However, only the results at a high temperature of 300 oC were provided in the text.

(8) The page for Ref. 47 was missed.

Response to Reviewers' Comments

REVIEWER #1

The authors have thoroughly revised their manuscript in response to reviewer comments, and is now suitable for publication. As a single point of feedback, I would encourage the authors to be more concise in future response letters; their response letter was 34 pages long! I appreciate the desire to be thorough but being more concise helps reviewers greatly.

Response: We thank the reviewer's positive endorsement of our revised version and recommendation of publication. We appreciate your time and effort in providing constructive comments and suggestions. We will try our best to make our response letter more concise in the future.

REVIEWER #2

In this work, DAST@cyclodextrin (HP β CD) host-guest supramolecular complex with strong fluorescence and high stability as well as some applications of the materials were reported. The authors have responded to all the comments of the reviewers one-by-one, and a major revision has been made. Most of their explanations are acceptable, at least in their cases. So, it seems that the possible publication could be considered. Before doing so, a minor revision is required.

Response: We appreciated the reviewer's positive recognition of our revision. Accordingly, we continued to thoroughly revise the manuscript based on the reviewer's constructive comments.

Comment #1. *About the copolymerization triggered by H₂O: Page 9 and Fig. 2c and d, the authors claimed that this phenomenon is attributed to "the hydrogen bonding interactions and copolymerization process triggered in the presence of H₂O". In contrast, "post-annealing treatment will block the copolymerization". To convince the readers, the authors can provide the proof(s) for their claims.*

Response: We thank the reviewer's constructive advice. Accordingly, we have provided an SEM image showing the morphology of DAST@HP β CD fibers after post-annealing treatment and exposure in high humidity condition, which maintained its well-interconnected fibrous morphology (Supplementary Fig. 9). We speculate that post-annealing treatment may inhibit the hydrogen bonding interactions triggered in the outer hydrophilic surface of HP β CD, thus preventing the copolymerization and secondary assembly between adjacent HP β CD molecules. The relevant discussion has been added to the revised manuscript.

As can be seen on Page 9, "Notably, the post-annealing treatment of DAST@HP β CD fibrous membrane will block the copolymerization process, and in this case, the individual fibers will not crosslink and merge together to form the compact thin film (Supplementary Fig. 9).

Supplementary Fig. 9. The SEM image of DAST@HP β CD fibers after post-annealing treatment and exposure in high humidity condition.

Comment #2. *About the TEM image: 1) Whether can DAST and HP β CD be determined by TEM? If yes, please show these two components in Fig. 2f, respectively; 2) It was described on page 9 “single-crystalline DAST crystals (i.e. \sim 12 nm in diameter)” in Fig. 2f. In this case, how DAST was encapsulated in the CD cavity?*

Response: We thank the reviewer for pointing this out and raising the insightful questions. 1) To be honest, it is quite difficult for us to exactly determine the amorphous HP β CD matrix in Fig. 2f, although one could faintly observe some amorphous regions around the highly crystalline composited crystals. **However, we could not exactly tell whether such amorphous regions were from HP β CD matrix or the carbon-based TEM grid.** In addition, the amorphous organic compounds can not tolerate the high energy e-beam irradiation for a long time during TEM measurement, which raised the difficulty of directly recognizing the amorphous HP β CD matrix. 2) We apologized for the incorrect statement of “single-crystalline DAST crystals (i.e. \sim 12 nm in diameter)”. Indeed, the observed nanocrystals in Fig. 2f should be ascribed to the “**DAST@HP β CD composited crystals**”, rather than the “pure DAST crystals”. Note, the analysis of the insertion of DAST into HP β CD cavity was based on the model of using a single molecule, as is evidenced by ICD signal, ^1H NMR and COSY NMR results. We found

that the lattice fringe of (121) plane of DAST@HP β CD composited nanocrystals (in this work) was slightly larger than those of previously reported DAST nanocrystals without any chemical modification (i.e. a lattice fringe of 0.445 nm for (121) plane (*Nanoscale* 8, 18882-18886 (2016)). The expansion of lattice fringe can be explained by the strong host-guest inclusion interaction between DAST and HP β CD. Accordingly, we have modified this statement in the revised manuscript.

As can be seen on Page 9, “The transmission electron microscope (TEM) characterizations revealed the morphology of nano-sized, crystalline DAST@HP β CD composited crystals (i.e. ~12 nm in diameter). We identified the well-resolved lattice fringes of 0.482 nm and 0.298 nm that could be indexed to the (121) and (006) crystal planes of DAST materials (Fig. 2f and Supplementary Table 4). We found that the lattice fringe of (121) plane of DAST@HP β CD composited nanocrystals was slightly larger than those of previously reported DAST nanocrystals without any chemical modification (i.e. a lattice fringe of 0.445 nm for (121) plane)¹⁰, which can be attributed to the strong host-guest inclusion interaction between DAST and HP β CD.”

Comment #3. *As summarized in Table 1, all the optical properties for the DAST@HP β CD fiber and DAST@HP β CD thin film are different, e.g. absorption and PL peaks, Eg, PLQY, etc. This is interesting since both are DAST@HP β CD. Why so?*

Response: We thank the reviewer for raising such an interesting question. Since the DAST@HP β CD thin film is fabricated under the high humidity conditions, it can be regarded as a kind of hydrated DAST-based supramolecular membrane, which would change the molecular orientation and arrangement (*Appl. Phys. Lett.* 75, 3291 (1999)), thus causing the shifts in both absorption and PL peaks, as well as the change of Eg, relative to the DAST@HP β CD fiber.

As for the different PLQYs, they can be related to the different morphologies with different light utilization capability. Typically, DAST@HP β CD thin film with dense morphology showed relatively inferior light absorption than that of the DAST@HP β CD fiber with porous morphology that could prolong the light utilization pathway (the absorption proportion of the thin film is 0.437, and the absorption

proportion of the fiber is 0.472, Supplementary Fig. 15 and 17). According to the equation of (1) (Adv. Mater. 9, 230–232 (1997)), one could determine the increased PLQY for the DAST@HP β CD thin film. The relevant discussion has been added to the revised manuscript.

$$\eta = \frac{\text{number of photons emitted}}{\text{number of photons absorbed}} \quad (1)$$

As can be seen on Page 13, “The different optical properties (i.e. absorption and PL peaks, Eg, PLQY, etc.) for the DAST@HP β CD fiber and DAST@HP β CD thin film can be attributed to their distinct differences in terms of hydration degree⁵⁶ and morphologies.”

Supplementary Fig. 15. The PLQY measurement of DAST@HP β CD fibers.

Supplementary Fig. 17. The PLQY measurement of DAST@HPβCD thin film.

Comment #4. (4) *About the thermotolerance: Supplementary Fig. 22f shows a black region on the left, whether some of the DAST@HPβCD thin film had been destroyed at 300 °C ?*

Response: We greatly appreciated the reviewer’s careful review and for raising this interesting question. The black region on the left of the DAST@HPβCD sample can be explained by the following facts. On one hand, considering the melting point of DAST is 261 °C (Supplementary Fig. 21), one could expect some degree of decomposition for DAST@HPβCD sample when it was heated at 300 °C for a while. Though, the resultant DAST@HPβCD film is still luminescent (Supplementary Fig. 23f). On the other hand, under high thermal stress at 300 °C, one could observe some regions of the film are curved and distorted, especially at the surrounding edges, which will affect the incident angles of UV illumination and cause the chromatism of observed luminescence. Accordingly, to more accurately describe the thermal stability of the DAST@HPβCD thin film, we have modified the description of thermotolerance of the film from 350°C to 300 °C.

As can be seen on Page 2, “The DAST@HPβCD membrane can continuously emit luminescence even being heated at 300 °C.....”

As can be seen on Page 4, “Due to high chemical and thermal stability of HPβCD hosts, the DAST@HPβCD membranes exhibited exceptional thermotolerance up to 300 °C....”

As can be seen on Page 25, “The 1PEF luminescence performance has been improved by 81-fold, and a record high PLQY of 73.5% has been achieved. The DAST@HPβCD thin film is thermotolerant even at 300 °C.....”

Comment #5. *About the Raman spectra in Supplementary Fig. 24: Why the peaks for DAST disappeared in H₂O, especially, for the peak at ~1190 nm?*

Response: We thank the reviewer for raising such an interesting question. The reason why the peaks can not be detected for the DAST dissolved in H₂O can be attributed to the trans–cis isomerization of stilbazolium. The peaks at 1180 cm⁻¹ (in-plane aromatic ring deformations, typically shown for para-substituted benzenes) and 1577 cm⁻¹ (typical C=C=C in-plane stretching frequency from carbon backbone between the aromatic rings of the stilbazolium chromophore) were normally shown in DAST crystal with **trans-state configuration** (*Figure R1a, Phys. Rev. B 66, 205107 (2002)*). When dissolved in water, DAST will easily isomerize to **cis-state configuration** (*Figure R1b, Chem. Rev. 103, 3899-4032 (2003); Nanoscale 8, 18882-18886 (2016)*). In this case, the symmetry of the DAST molecule will alter, which will change the deformation vibration mode of the dimethylaminophenyl group. In water solution, relative to the trans-state analogues, the cis-state DAST molecules will account for a considerable proportion, and their contribution to the peaks at abovementioned 1180 and 1577 cm⁻¹ would be largely weakened, and eventually, the signals of DAST can be easily overshadowed by the Raman peaks of water. In contrast, the DAST@HPβCD dissolved in water solution showed similar peaks at 1188 and 1592 cm⁻¹, respectively, suggesting the HPβCD inclusion can largely maintain the trans-state configuration of DAST molecules. Accordingly, we have further explained this in the revised manuscript.

As can be seen on Page 19, “Raman spectra showed that, in water environment, only when the DAST is included by HPβCD, in which the trans–cis isomerization of stilbazolium is largely suppressed, its characteristic peaks preserve and show up

(Supplementary Fig. 25).”

Figure. R1. The trans-state (a) and cis-state (b) of DAST molecule. Note, the tosylate anion has been omitted for better clarity.

Comment #6. For the one-photon-excited emission, two-photon excited fluorescence (2PEF), and three-photon absorption (3PA), whether the differences between DAST and DAST@HP β CD can be summarized?

Response: We thank the reviewer’s constructive advice. According to your suggestion, we summarized the differences of 1PEF, 2PEF and 3PEF properties between DAST and DAST@HP β CD, as can be seen in Supplementary Table 5.

As can be seen on Page 13, “Accordingly, we summarized the differences of 1PEF, 2PEF and 3PEF properties between DAST and DAST@HP β CD, as can be seen in Supplementary Table 5.”

Supplementary Table S5. Summarized 1PEF, 2PEF and 3PEF properties of DAST(powder/nanocrystal) and DAST@HP β CD fibers.

Material	1PEF Emission/Excitation (nm)	2PEF Emission/Excitation (nm)	3PEF Emission/Excitation (nm)
DAST	615, 746 / 420 ^a (powder)	606/932 ^[1] 585/800 ^[2] (nanocrystal)	N/A
DAST@HP β CD fibers (this work)	583/450 ^b	578 / 770 ^c 584 / 1000 ^d	585/1590 ^e

^{a, b} The data is extracted from Supplementary Fig. 13.

^{c, d} The data is extracted from Fig. 5b.

^eThe data is extracted from Fig. 5c.

[1] Tian, T., Cai, B. & Sugihara, O. DAST single-nanometer crystal preparation using a substrate-supported rapid evaporation crystallization method. *Nanoscale* **8**, 18882-18886 (2016).

[2] Zheng, Mei-Ling. *et al*, Two-photon excited fluorescence and second-harmonic generation of the DAST organic nanocrystals. *J. Phys. Chem. C* **115**, 8988-8993 (2011).

Comment #7. *It was described in both abstract and conclusion that the DAST@HP β CD thin film is stable even at 350 °C. However, only the results at a high temperature of 300 °C were provided in the text.*

Response: We appreciated the reviewer's careful review and pointing this out. Accordingly, we have modified the relevant statement in the revised manuscript.

As can be seen on Page 2, "The DAST@HP β CD membrane can continuously emit luminescence even being heated at 300 °C....."

As can be seen on Page 4, "Due to high chemical and thermal stability of HP β CD hosts, the DAST@HP β CD membranes exhibited exceptional thermotolerance up to 300 °C...."

As can be seen on Page 26, "The 1PEF luminescence performance has been improved by 81-fold, and a record high PLQY of 73.5% has been achieved. The DAST@HP β CD thin film is thermotolerant even at 300 °C....."

Comment #8. *The page for Ref. 47 was missed.*

Response: We highly appreciate the reviewer's careful review and picking up the errors. Accordingly, we have added the page for Ref.47 in the revised manuscript.

*As can be seen on Page 30, "47 Chen, L. et al. Reversible emitting anti-counterfeiting ink prepared by anthraquinone-modified β -cyclodextrin supramolecular polymer. *Adv. Sci.* **7**, 2000803 (2020)."*

REVIEWERS' COMMENTS

Reviewer #2 (Remarks to the Author):

The authors have responded to all the comments of the reviewers one-by-one, and the related revisions have been made. Now, it could be considered for the possible publication in Nature Communications.